# The epichaperome is a mediator of toxic hippocampal stress and leads to protein connectivity-based dysfunction

Maria Carmen Inda[1,17,19], Suhasini Joshi[1,19], Tai Wang[1,19], Alexander Bolaender[1,19], Srinivasa Gandu[1], John Koren III[1], Alicia Yue Che[2], Tony Taldone[1], Pengrong Yan[1], Weilin Sun[1], Mohammad Uddin[1], Palak Panchal[1], Matthew Riolo[1], Smit Shah[1], Afsar Barlas[3], Ke Xu[3], Lon Yin L. Chan[1], Alexandra Gruzinova[1], Sarah Kishinevsky[1,4], Lorenz Studer[4], Valentina Fossati[5], Scott A. Noggle[5], Julie R. White[6], Elisa de Stanchina[7], Sonia Sequeira[8], Kyle H. Anthoney[9], John W. Steele[9], Katia Manova-Todorova[3], Sujata Patil[10], Mark P. Dunphy[11], NagaVaraKishore Pillarsetty[11], Ana C. Pereira[12,18], Hediye Erdjument-Bromage[13,14], Thomas A. Neubert[13,14], Anna Rodina[1], Stephen D. Ginsberg[15,16], Natalia De Marco Garcia[2], Wenjie Luo[2,20]* & Gabriela Chiosis[1,20]*

Optimal functioning of neuronal networks is critical to the complex cognitive processes of memory and executive function that deteriorate in Alzheimer's disease (AD). Here we use cellular and animal models as well as human biospecimens to show that AD-related stressors mediate global disturbances in dynamic intra- and inter-neuronal networks through pathologic rewiring of the chaperome system into epichaperomes. These structures provide the backbone upon which proteome-wide connectivity, and in turn, protein networks become disturbed and ultimately dysfunctional. We introduce the term protein connectivity-based dysfunction (PCBD) to define this mechanism. Among most sensitive to PCBD are pathways with key roles in synaptic plasticity. We show at cellular and target organ levels that network connectivity and functional imbalances revert to normal levels upon epichaperome inhibition. In conclusion, we provide proof-of-principle to propose AD is a PCBDopathy, a disease of proteome-wide connectivity defects mediated by maladaptive epichaperomes.

[1] Chemical Biology Program, Memorial Sloan Kettering Cancer Center, New York, NY 10065, USA. [2] Brain and Mind Research Institute, Weill Cornell Medical College, New York, NY 10065, USA. [3] Molecular Cytology Core Facility, Memorial Sloan-Kettering Cancer Center, New York, NY 10065, USA. [4] Department of Developmental Biology, Memorial Sloan Kettering Cancer Center, New York, NY 10065, USA. [5] The New York Stem Cell Foundation Research Institute, New York, NY 10019, USA. [6] Comparative Pathology Laboratory, Memorial Sloan Kettering Cancer Center, New York, NY 10065, USA. [7] Molecular Pharmacology Program, Memorial Sloan Kettering Cancer Center, New York, NY 10065, USA. [8] Office of Clinical Research, Memorial Sloan Kettering Cancer Center, New York, NY 10065, USA. [9] Department of Biological Sciences, Humboldt State University, Arcata, CA 95521, USA. [10] Department of Epidemiology-Biostatistics, Memorial Sloan Kettering Cancer Center, New York, NY 10065, USA. [11] Department of Radiology, Memorial Sloan Kettering Cancer Center, New York, NY 10065, USA. [12] Department of Neuroscience, Rockefeller University, New York, NY 10065, USA. [13] Department of Cell Biology, NYU School of Medicine, New York, NY 10016, USA. [14] Kimmel Center for Biology and Medicine at the Skirball Institute, NYU School of Medicine, New York, NY 10016, USA. [15] Departments of Psychiatry, Neuroscience & Physiology & the NYU Neuroscience Institute, NYU School of Medicine, New York, NY 10016, USA. [16] Center for Dementia Research, Nathan Kline Institute, Orangeburg, NY 10962, USA. [17] Present address: Hostos Community College, City University of New York, The Bronx, NY 10451, USA. [18] Present address: Icahn School of Medicine at Mount Sinai, New York, NY 10029, USA. [19] These authors contributed equally: Maria Carmen Inda, Suhasini Joshi, Tai Wang, Alexander Bolaender. [20] These authors jointly supervised this work: Wenjie Luo, Gabriela Chiosis. *email: wel2009@med.cornell.edu; chiosisg@mskcc.org

The hippocampus, a brain area involved in forming and storing memories, is one of the earliest structures to undergo neurodegeneration when exposed to chronic stress[1]. Mild, acute stress often enhances hippocampal function by augmenting synaptic plasticity, reflecting the adaptive importance of remembering threatening or dangerous circumstances[2]. However, these same mechanisms, when activated intensely or for a prolonged period, may render the hippocampus susceptible to detrimental effects of chronic stress, ultimately leading to synaptic failure and cognitive deficits[3–5]. In Alzheimer's disease (AD), the most prevalent form of neurodegeneration, stressors such as aging, genetics, environmental factors, and others, accumulate. These result in imbalances in the connectivity of neuronal circuitry, and also negatively impact the intracellular connectivity of neuronal proteins and protein pathways, contributing to cognitive decline associated with AD[6]. Current technical limitations curb our ability to explore global changes in protein level connectivity, restricting how to translate key network changes into AD therapies. They also challenge our ability to diagnose disorders early in the disease process where therapeutic intervention would be most effective.

Chaperones have long been associated with the management of cellular stress, including neuronal stressors[7–11]. Changes in chaperone expression are believed to be at the core of many diseases, including neurodegenerative disorders where proteotoxicity occurs due to protein misfolding and aggregation[7–9]. Chaperones' effects are executed in a one-on-one, cyclic fashion, aiding protein folding, degradation or disaggregation[12]. Pertinent to AD pathophysiology, heat shock protein 90 (HSP90)/co-chaperone complex folds tau or hyperphosphorylated tau, whereas heat shock protein 70-carboxyl-terminus of HSP70 Interacting protein (HSP70-CHIP) complex mediates degradation[13,14]. Both inhibition of chaperones supporting accumulation of hyperphosphorylated species (e.g., HSP90) and induction of chaperones promoting tau degradation (e.g., HSP70) have been proposed for AD treatment[15]. HSP90 inhibition may also enhance synaptic protein expression via activation of heat shock factor 1 (HSF-1) transcriptional regulation[16].

We recently showed chronic stressors resculpt chaperone machinery and function[17–21]. Within a cancer paradigm, chronic stresses associated with oncogenic transformation may increase connectivity among chaperones, co-chaperones, and other factors collectively referred to as the 'chaperome'. This is accomplished by increasing both the interaction strength and number of interactions among participant proteins, which is not necessarily accompanied by a change in expression levels. Termed 'epichaperomes', these structurally modified chaperome pools do not act in protein folding and degradation per se, but rather as multi-molecular scaffolds that pathologically remodel cellular processes[17,18,20,22]. This mediates alterations in the proteome associated with disease[17,18,20,23–25]. How this maladaptive switch of the chaperome into epichaperomes affects organization and function at proteome-wide level, and whether these effects reverberate to the level of target organ dysfunction remains unknown.

We show a switch from chaperome into epichaperomes in AD alters connectivity of the neuronal proteome leading to network-wide dysfunction and cognitive decline (Supplementary Fig. 1A). We propose that functionally AD is a protein connectivity-based dysfunction (PCBD), a previously unappreciated aspect of AD biology. We uncover a novel mechanism that reconfigures the chaperome to epichaperome within vulnerable brain regions that potentially underlies neurodegenerative disorders with significant stress-related pathobiology, including AD. We provide evidence how protein connectivity dysfunctions alter protein-to-neuronal circuit-to-organ level changes. We show that these epichaperome-mediated intra- and inter-neuronal network-wide disturbances are identifiable and targetable. We used innovative chemical biology tools and unbiased functional proteomics-bioinformatics pipelines recently developed for epichaperome study in cell-stress paradigms[17,20,21,24,25] (Supplementary Fig. 1B), and applied them to tauopathies as a model of molecular stress and human AD brains.

## Results

**Epichaperomes in the murine brain.** We assessed a switch of the chaperome into epichaperomes in a regional and age-dependent manner within tau transgenic mice (Fig. 1 and Supplementary Fig. 2). We used PS19 mice which present with hippocampal synaptic loss and dysfunction that are detected before tau tangles emerge[26]. These mice also display alterations associated with AD, including spatial learning and memory deficits and a disinhibition phenotype[27]. For comparison, we used age-matched wild type (WT) mice.

We observed in PS19 mice of 3 to 7–8 months of age epichaperomes were exquisitely abundant in brain regions associated with memory and learning. Aberrant epichaperomes were found within the hippocampal formation (HPF), including the entorhinal cortex (ENT), hippocampus (HIP), and subiculum (SUB), as well as the amygdala (AMG), but were largely missing or low in other brain regions (Fig. 1a–c). The highest presence at this age was in the dentate gyrus (DG) and hippocampal CA3 layer (CA3). Strong epichaperome staining was assigned to the granule cell (DG-sg), molecular (DG-mo), and polymorph (DG-po) DG layers. In CA3 the epichaperome was abundant in several laminae including the pyramidal layer (CA3sp), stratum radiatum (CA3sr), stratum lucidum (CA3slu), and stratum oriens (CA3so). The pyramidal layer of CA1 (CA1sp) showed highest epichaperome expression in the CA1 region, although we also noted its presence in stratum radiatum (CA1sr) and stratum lacunosum-moleculare (CA1slm). Similar epichaperome expression was found in the dorsal and ventral HPF (Fig. 1b). In neocortex, epichaperome expression was highest in temporal lobe areas adjacent to the HPF and AMG, with a pattern of spreading towards somatosensory areas (Fig. 1c).

Epichaperome expression in the PS19 hippocampus displayed a linear increase between 3 and 8 months of age, followed by a plateau (Fig. 1d, e). With increasing age, epichaperome staining became less restricted to the HPF. By 11 months, epichaperome analysis in PS19 mice revealed its presence in most brain areas, and similarly to the HIP, its expression also reached a maximal value in these areas, albeit at a later timepoint (Fig. 1e). Notably, the HPF retained the highest level of epichaperome formation at all ages, commensurate with the hypothesized nidus of human AD pathology. In age-paired WT mice, we detected no or little epichaperomes (Fig. 1d, e). Individual chaperones and co-chaperones including HSP90, HSP70 and HSP110 isoforms, key building blocks for the epichaperome networks[17,21], were expressed at similar levels throughout the brain, and not significantly different between PS19 and WT mice (Fig. 1f and Supplementary Fig. 2A), similar to observations in cancer[17].

**The epichaperome and tau pathology.** In PS19 mice, hippocampal synaptic loss and dysfunction is linked to tau toxicity prior to tau tangles[26]. Similarly, in AD, spreading of tau pathology may induce synaptic destruction followed by axonal and later somatodendritic degeneration[6]. Whereas tau toxicity remains debatable, hyperphosphorylation of tau is involved in spreading tau pathology[28]. Analyses on both human[29] and PS19 mice[30] found immunohistochemical staining for hyperphosphorylated

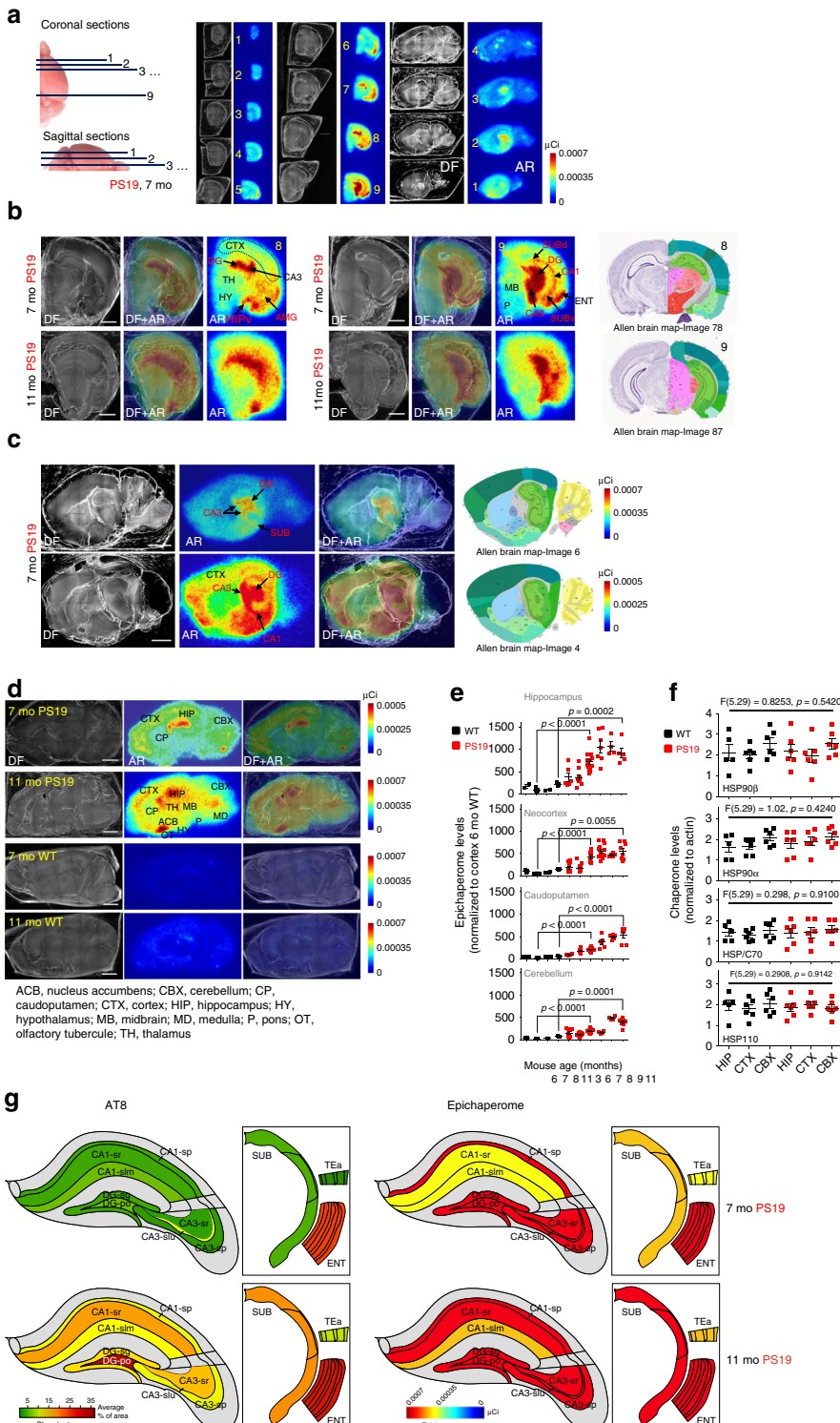

**Fig. 1 The epichaperome is expressed in the PS19 mouse brain in a region- and age-dependent manner. a** Sectioning and staining to map epichaperome network localization via autoradiography (AR) using [131]I-PU-AD, a radiolabelled chemical probe for epichaperome detection and quantification, is shown for a representative mouse brain (PS19, 7 months). Dark field (DF) microscopy images were taken to visualize brain anatomy. **b–d** Indicated coronal (**b**) and sagittal (**c**, **d**) brain sections like in (**a**) show epichaperome localization and its distribution in PS19 (n = 5), and its absence in WT (n = 5), mouse brains. Scale bar, 2 mm. Images for anatomical location of brain areas closest to those used for epichaperome detection were obtained from the Allen Institute for Brain Science, Mouse, P56. **e** Epichaperome levels analyzed as in (**a**) in the indicated brain regions in WT and age-matched PS19 mice show its age-dependent and PS19-specific expression. Graph, mean ± SEM; unpaired two-tailed t-test, PS19 (n = 14 and 5) vs WT (n = 7 and 5) at 7 and 11 months, respectively. **f** Levels of individual chaperones in 8–10 months old WT (n = 6) and PS19 (n = 6) mice, determined by western blot. Graph, mean ± SEM; one-way ANOVA. See also Supplementary Fig. 2A. **g** Schematic summary showing the spatio-temporal expression of the epichaperome compared to hyperphosphorylated human tau (measured by AT8, IHC, see Supplementary Fig. 2B) in PS19 mice. Source data are provided as a Source Data file.

tau (AT8; recognizing Ser202/Thr205) to correlate with tau pathology progression.

We found the switch of the chaperome into epichaperomes precedes tau pathology in the PS19 hippocampus (Fig. 1g). While the epichaperome was already evident in the HIP at 3 months and reached maximal levels by 7–8 months in PS19 mice, AT8 staining was not detected at 3 months[27]. Moreover, minimal AT8 HIP signal was evident at 7 months when epichaperome staining has fully encompassed the HIP (Fig. 1g and Supplementary Fig. 2B). Epichaperome inhibition reduced soluble tau, hyperphosphorylated tau (AT8), and some oligomeric tau species in PS19 mice (Supplementary Fig. 2B–D). Therefore, epichaperomes may form before tau pathology, and support processes that enable tau pathology formation.

**Epichaperomes in the human brain**. We confirmed the presence of the epichaperome in human postmortem brains (Fig. 2a–e) and in live patients through epichaperome imaging, providing proof-of-principle (Supplementary Fig. 3A–C).

In healthy cells, chaperones have evolved to interact with proteins in a dynamic manner[12]. Dynamic protein complexes disassemble under native protein chromatography and chaperones (e.g., HSP90 and HSC70) are detected as homodimers (Fig. 2b, c, blue arrow and refs. [17,21]). In contrast, when incorporated into epichaperome networks, interactions between chaperome units become stabilized and interaction partners change[17,21]. Epichaperomes are retained under native PAGE conditions and appear as multiple high-molecular weight complexes (Fig. 2a and refs. [17,21]).

We found human AD brains, but not age-matched non-demented (ND) brains, contained a significantly higher fraction of the total chaperome rewired into epichaperomes (see Fig. 2b and Supplementary Fig. 3D for epichaperome network hubs such as HSP90, HSP-organizing protein (HOP) and HSC70)[17,21], and demonstrated a pattern of distinct interactions (see Fig. 2b native PAGE for HSP90, HSC70, HOP and CDC37). In a protein–protein interaction (PPI) network, hubs are a small number of highly connected nodes (i.e., proteins).

As in cancer cells[17], AD epichaperomes contained hub proteins, such as HSP90, HSC70, HSP110, HOP, and CDC37 (Fig. 2b, c), independent of the total expression of these individual chaperones (Fig. 2b, f and Supplementary Fig. 3D), and unlike the folding chaperones HSP90 and HSP70[13,14], epichaperomes in the AD brain did not bind to hyperphosphorylated tau or to full-length APP (Fig. 2d). The epichaperome networks were sensitive to inhibitors such as PU-H71 and PU-AD (Fig. 2c, f). Binding by these inhibitors to HSP90 residing in epichaperome networks resulted in an initial trapping and subsequent collapse of epichaperome-mediated networks, without affecting the total levels of individual chaperones (Fig. 2f).

**Epichaperomes mediate PCBD in AD**. We investigated epichaperome composition and function in AD through unbiased proteome-wide chaperomics (Fig. 3a–d). We used an innovative chemical proteomics approach where epichaperome affinity-purification baits (i.e., PU-beads)[17,21] capture the epichaperome and its interactome. Cargo is submitted for unbiased mass spectrometry (MS) analysis to identify and quantify proteins that are present. We performed proteome capture with excess bait to enable the acquisition of physiological chaperome–proteome interactions[17,21,24] in addition to those established by the epichaperome in AD. We applied the platform to human brains (AD vs ND), iPSC-derived neurons (*APP* duplication vs WT), transgenic mouse brains (PS19 vs WT), and cellular models of human

tau toxicity (N2a cells overexpressing human tau vs N2a cells with vector only) (Fig. 3a).

We analysed global PPI networks in AD and ND brain samples and found a large change in protein connectivity executed through rewiring of the chaperome into epichaperomes, with approximately 62% of the nodes and 98.5% of the edges significantly altered in AD (Fig. 3b). When compared to the inherent physiological organization of the proteome in ND brains, new connections were formed in AD (aberrant changes in 35% of nodes and 43% of edges) whereas many normal connections were lost (aberrant changes in 27% of nodes and 55.5% of edges) (Fig. 3b).

To understand the relationship between PPI connectivity changes and their functional significance, we implemented an interactome gene-set enrichment analysis (iGSEA) over biological processes important for synapse biology (curated by Synapse Gene Ontology and Annotation Initiative, SynGO). We found the chaperome in ND brains regulated housekeeping proteostasis through its known roles in protein transport[31] (see axonal transport) and in de novo protein folding during polypeptide synthesis {see regulation of translation and connectivity to chaperones such as HSP70s, CH60 (HSP60), HSPB1, HSP90s, AHSA1, TERA (VCP)} (Fig. 3c, blue lines).

Conversely in AD brains, structural rewiring of the chaperome into epichaperomes resulted in a functional switch where the functional focus of the epichaperome was on regulation of glutamatergic synaptic transmission, regulation of AMPA receptor activity, regulation of postsynaptic membrane potential, modulation of synaptic transmission, regulation of long-term neuronal synaptic plasticity, negative regulation of neurotransmitter, negative regulation of protein secretion and synapse assembly and organization (Fig. 3c, red lines). These processes are related to synapse formation and function, long-term potentiation (LTP), and learning (Fig. 3c, red lines), suggesting the switch of the chaperome into epichaperomes has an important functional role, including synaptic dysfunction in AD.

We also found that functions seemingly preserved between chaperome and epichaperome were in fact executed by distinct subsets (Fig. 3c). For example, regulation of translation was associated with two different subsets of chaperome members in ND versus AD, which may signify that in AD there is a shift away from de novo protein folding, a function of the chaperome under physiological conditions, which is a novel observation driven by this protein network assessment methodology.

We analyzed the complement of protein pathways dysregulated through the switch of the chaperome into epichaperomes, providing a mechanistic link between epichaperome-mediated proteome-wide connectivity change and functional proteome-wide disturbances in AD. Commensurate with connectivity rewiring in PPI networks effected by the epichaperome, we observed pathway alterations at the global proteome level (Fig. 3d). Despite proteome variability characteristic to each patient's disease trajectory in AD cases (PCA plot, Fig. 3d), we found proteomes manifested common epichaperome-mediated defects. Epichaperome-dependent protein pathways include networks important for synaptic plasticity, cell-to-cell communication, protein translation, cell cycle re-entry, axon guidance, metabolic processes and inflammation, among others (Reactome pathway analyses, Fig. 3d).

While each network warrants further in-depth investigation, we focused initially on protein pathways important for synaptic function and memory, functionalities clearly impaired during the onset and progression of dementia. Short-term memories rely on post-translational modification of pre-existing synaptic proteins, whereas more persistent memories require the production of new proteins[32–39]. De novo protein synthesis within neurons,

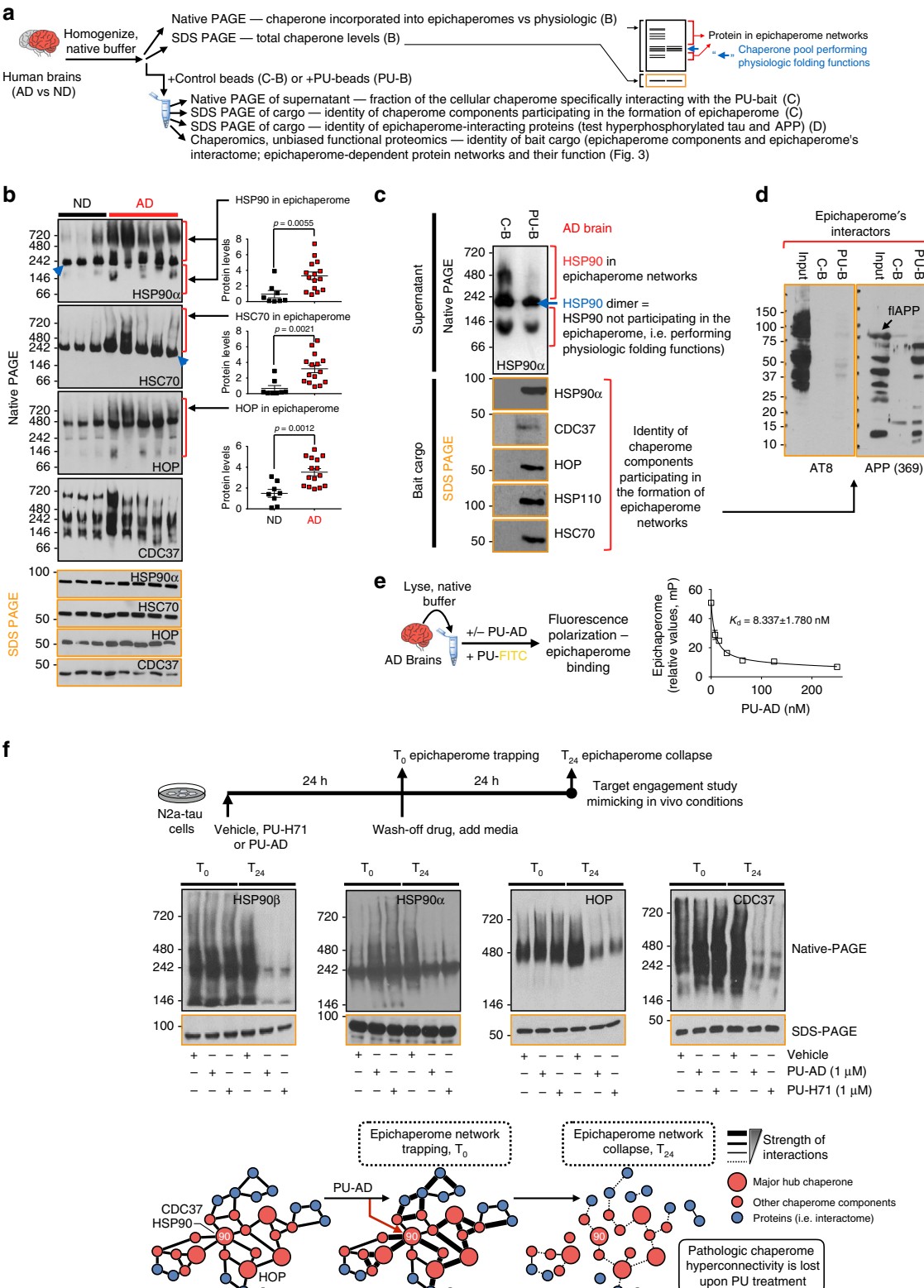

**Fig. 2 Epichaperome expression is specific to the AD brain. a** Schematic showing methods to differentiate normal chaperome pools from epichaperome networks (i.e. with function in pathologic PPI network connectivity). **b** Biochemical signature of major chaperone proteins in AD brains and normal, age-matched, brains (non-demented, ND). Representative gel of $n = 15$ for AD and $n = 8$ for ND, is shown. Graph, mean ± SEM, unpaired two-tailed $t$-test. See also Supplementary Fig. 3. **c**, **d** Affinity-purification of the epichaperome and its AD interactome shows no specific interaction between the epichaperome and hyperphosphorylated tau or full-length APP (flAPP). Data representative of $n = 3$ individual AD brains. PU-bait (PU-B); control bait (C-B). **e** Binding of PU-AD to the epichaperome in AD brain homogenates. Graph, mean ± SEM, $n = 3$ replicates. **f** PU-AD disrupts the epichaperome network. Schematic of the experimental design and the biochemical signature of select epichaperome network components are shown. Representative data of $n = 3$ independent experiments. Source data are provided as a Source Data file.

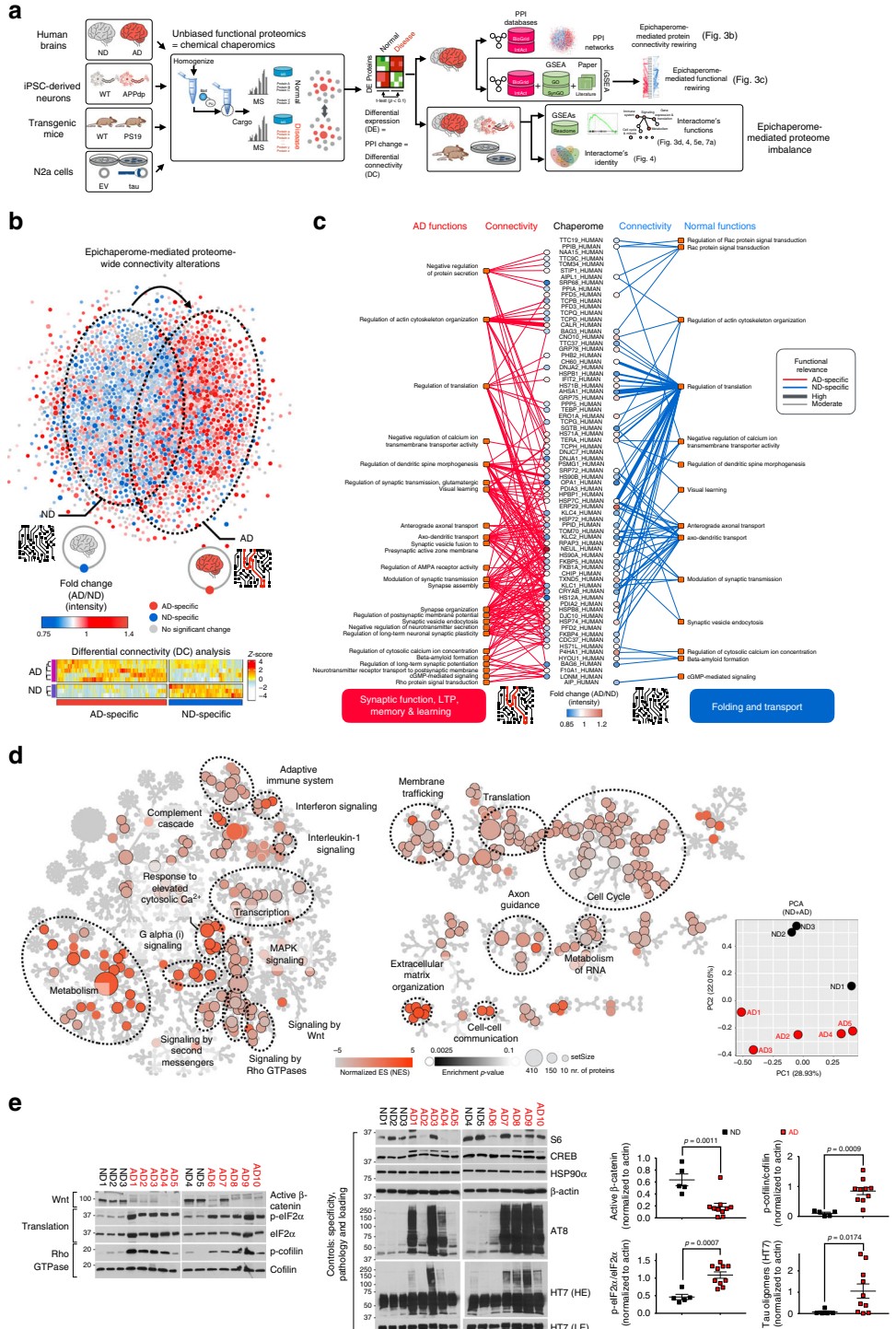

**Fig. 3 The epichaperome mediates pathologic changes in proteome connectivity and function in AD. a** Workflow used to identify epichaperome constituents and its interactome, and establish the identity and function of epichaperome-mediated protein–protein interaction (PPI) networks in AD. Individual AD (*n* = 5) and ND (*n* = 3) human brains were analysed in 2–4 replicates, and compared to indicated mouse and cellular models of AD. **b** PPI network map (top) and heatmap (bottom) showing epichaperome-mediated changes in the proteome and in protein connectivity. Each dot represents a protein (blue, ND-specific, *n* = 942; red, AD-specific, *n* = 1191). Edges (i.e., connections or PPIs) were removed for simplicity of presentation. Heatmap; differentially connected (DC) proteins, *p* < 0.1. DC = chaperome–proteome connectivity change. **c** Chaperome–proteome connectivity and its functional significance annotated over biological processes important for synapse biology. In the map's center are chaperome members that participate in rewiring the connectivity of synaptic proteins between ND and AD; each blue and red line represents a PPI, and its functional outcome is presented at the end of each connection. **d** Functional analysis of the epichaperome and its interactome through Reactome pathway analyses uncovers protein pathways altered in AD through the epichaperome-mediated mechanism. Dysregulations in these pathways are shared among the sporadic AD brains. Principal component analysis (PCA) shows little relatedness among the individual AD interactomes. **e** Key effectors of select synaptic protein pathways from (**d**) were confirmed by western blot. HE, LE, high or low exposure, respectively. Graph, mean ± SEM, unpaired two-tailed *t*-test, with Welch's correction for the tau oligomers, *n* = 5 for ND and *n* = 10 for AD. Source data are provided as a Source Data file.

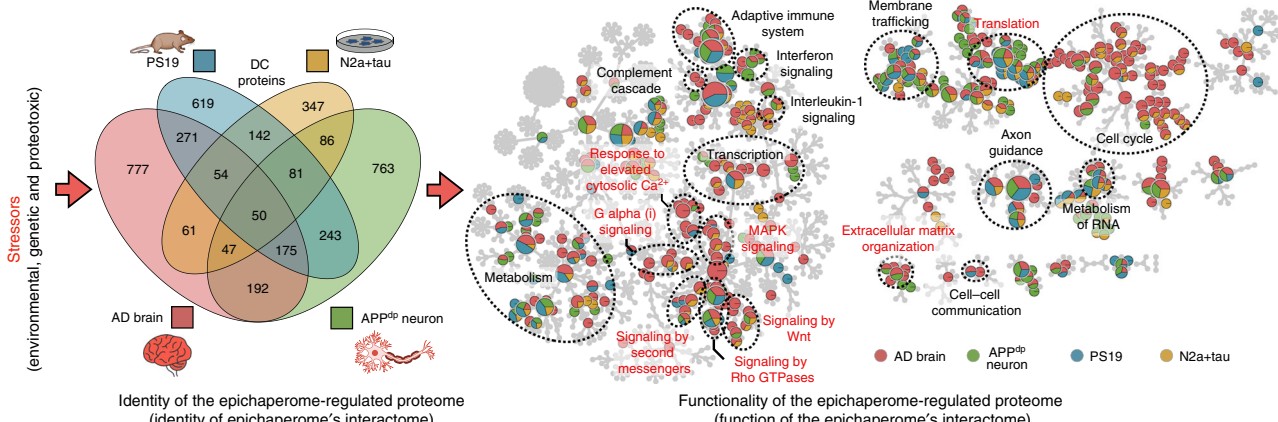

**Fig. 4 Epichaperome's interactome is stressor-specific, yet these distinct interactomes converge functionally on synaptic protein pathways.** The identity and function of the epichaperome and its interactome following the indicated neuronal stresses are shown in Venn diagram and pathway enrichment analysis representation, respectively. In the Venn diagram, each circle represents the number of proteins affected by the switch of the chaperome into epichaperome by a specific stressor condition. In the Reactome map, each pie (circle) represents a function (i.e., a protein pathway). If the circle is blue, yellow, red and green, all four stressors (or the stressors characteristic of each four conditions) induce imbalances in that specific protein pathway. If a circle is exclusively red, the pathway alteration is AD specific. The location of protein pathways with major roles in synaptic plasticity is denoted in red lettering. Other highly represented functional clusters are presented in black lettering. The interactive Cytoscape file associated with this figure provides the identity of each protein pathway and of proteins identified in each pathway. See also Fig. 5e.

underlying long-term synaptic plasticity, involves not only transcriptional regulation, but translational regulation[40]. Among the protein pathways important for the execution of these two phases are signalling networks such as signalling by second messenger, Galpha(i) signalling, signalling by Rho GTPases, signalling by Wnt, response to elevated cytosolic $Ca^{2+}$, MAPK signalling, adhesion-regulatory networks such as extracellular matrix organization and cell–cell communication, as well as protein translation-related networks, which our analyses indicate are dysregulated through the pathologic switch of the chaperome into epichaperomes (Fig. 3d). We validated biochemically that these pathways with key roles in synaptic fitness are indeed functionally imbalanced in AD (Fig. 3e). Specifically, we showed significant activity changes in AD when compared to ND brains by probing key effector proteins of these pathways. Among them are p-cofilin, an effector of Rho GTPase signalling important for actin remodelling and cytoskeletal reorganization[32,35,36], active β-catenin, an effector of Wnt signalling important for adhesion[36] and p-eIF2α, a protein translation effector indicating global translational activity repression[32,41,42].

In addition to the evaluated AD cases, each likely with a distinct underlying disease cause, we found overexpression of the human T40 isoform of tau (4R2N), introduction of an APP mutant (*APP* duplication) in neuronal cells, or overexpression of the human T34 isoform of tau (1N4R) with the P301S mutation in mouse brains (i.e., PS19 mice), to each promote functional imbalances in some, if not all, of these synaptic protein networks through the switch of the chaperome into epichaperomes (Fig. 4). This is important, as it may signify a common mechanism in neurodegenerative diseases associated with tau-induced and/or tau-associated stresses. Here, and despite largely distinct proteomes intrinsic to each patient's disease, a common functional denominator or common final path may be found where these proteomes manifest similar defects in protein networks important for synapse formation and neuroplasticity. Our results indicate that these stress-induced, dynamic protein network maladaptive alterations in synaptic protein pathways may be mediated through the switch of the chaperome into epichaperomes.

To test this hypothesis, we investigated the effect of tau overexpression as an example of a tau-related stressor in a

neuronal cell line (Fig. 5a, b). We confirmed that introduction of human tau was sufficient to rewire a fraction of the cellular chaperome into epichaperome networks (Fig. 5c, d) resulting in functional imbalances within synaptic protein networks which were also detected in human AD brains (Fig. 5e, f). Evidence was provided by fluorescence polarization where increased binding of the PU-FITC probe indicates greater incorporation of HSP90 into epichaperome networks (Fig. 5c)[17–21], by native-PAGE, where an increase in the number of stable epichaperome complexes was noted (Fig. 5d), and by western blot where the function or expression of several effector proteins involved in adhesion and actin remodelling, CREB activation, translation initiation and AMPAR phosphorylation[32–38,43] was imbalanced (Fig. 5f). Pharmacologic epichaperome inhibition by PU-AD rebalanced activity of networks to pre-tau overexpression states (Fig. 6a, b), and confirmed epichaperomes as mediators of tau-induced functional imbalances in synaptic pathways. Effects of PU-AD were independent of HSF-1 activation (Fig. 6c), distinguishing this mechanism of action from that observed with classical HSP90 inhibitors such as geldanamycin[16]. Importantly, PU-AD had no toxic effect on iPSC-derived AD neuronal cells in culture, even when human neurons where cultured for 72 h with PU-AD concentrations as high as 50 μM (Fig. 6d, e).

**PCBD and spatial memory impairment.** To link intra-neuronal network disturbances to target organ level dysfunction we investigated whether epichaperome inhibition by PU-AD reverts synaptic protein network imbalances in vivo, which were detected in PS19 mice via chemical chaperomics (Fig. 7a), to improve memory and learning[32–38]. Before using PU-AD in vivo, we established PU-AD has adequate bioavailability, epichaperome-specificity in vivo and a desirable pharmacokinetic profile that allows for extended therapeutic action at the target (Supplementary Fig. 4A–E). Target engagement was confirmed when PU-AD (75 mg kg$^{-1}$, i.p., three times weekly) productively engaged the epichaperome in PS19 mice (Fig. 7b). PU-AD delivery resulted in PCBD attenuation as evidenced by significant rebalancing activity of synaptic protein networks to levels observed in WT mice (Fig. 7c, d and Supplementary Fig. 5A, B).

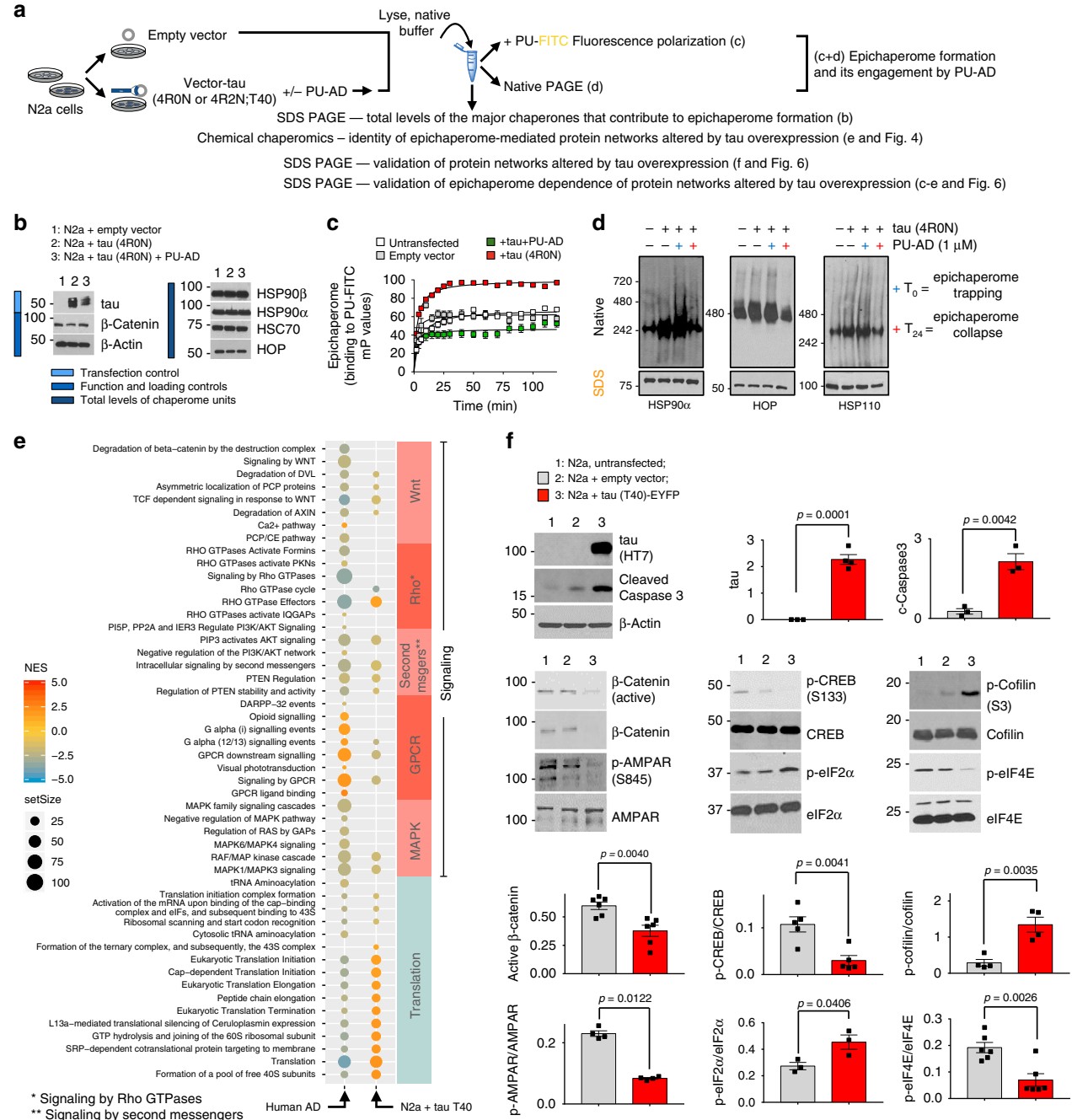

**Fig. 5 Tau overexpression causes epichaperome-mediated dysfunctions in synaptic protein pathways. a** Experimental design to investigate the effect of tau overexpression on the epichaperome network and on synaptic protein pathways. **b** Western blot, **c** fluorescence polarization and **d** native PAGE analyses confirm epichaperome network formation upon human tau introduction into N2a cells. **b, d** Gels are representative of three independent experiments. **c** Error bars show mean ± SEM, $n = 3$ independent experiments performed in triplicate. **e** Dot plot shows the identity of stress-specific dysfunctions in synaptic protein pathways that are mediated by the epichaperome, as identified by chaperomics, in human AD versus ND brains and in N2a cells expressing the T40 tau isoform versus empty vector. Select Reactome pathways with focus on those important for synaptic fitness are shown. Normalized enrichment score (NES) is indicated by node colour. The number of DC proteins (setSize) are indicated by the size of the nodes. Details on DC analyses and GSEA are described in Methods. **f** Western blot analyses validate the identity of pathways altered by tau through an epichaperome-mediated mechanism. Chosen for validation are: β-catenin degradation related to impaired axon-dendrite contact during synapse (see Wnt pathway); defective actin remodelling which we monitor via p-cofilin (see RhoGTPase and Galpha(12/13) signalling); defects in signalling networks that converge on CREB phosphorylation and activation of its transcriptional activity (see signalling by second messengers and MAPK pathways) and defective protein synthesis (see protein translation pathways). These pathway dysregulations converge on a defective LTP; phosphorylation of the GluA1 subunit of AMPAR on S845 is a critical step in LTP, and we use it as a biochemical LTP surrogate. Graph, mean ± SEM, unpaired two-tailed t-test, $n = 3–6$, as indicated. Representative blots are also shown. Source data are provided as a Source Data file.

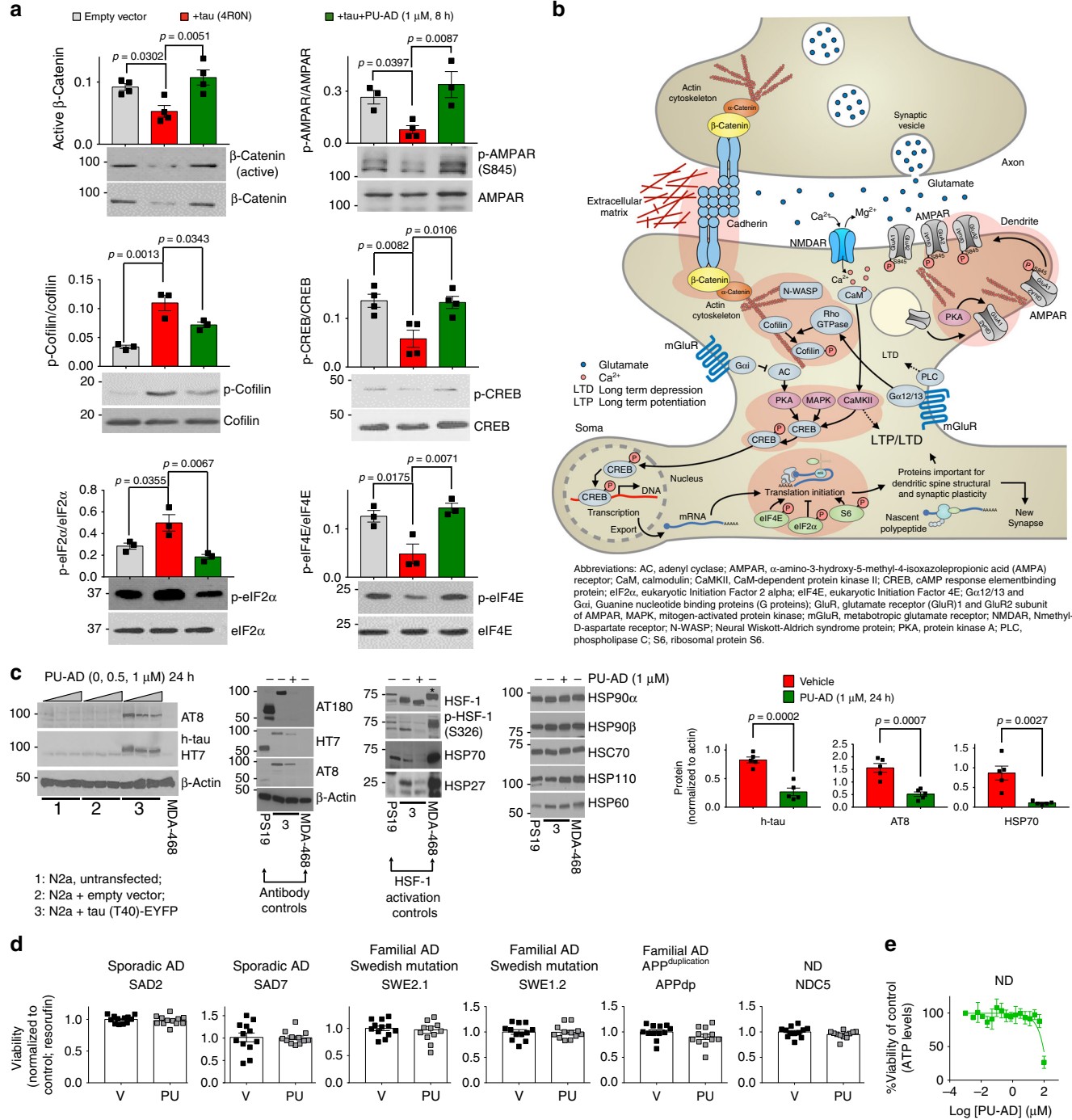

**Fig. 6 PU-AD de-connects epichaperome networks and reverts tau's negative effect on protein pathways important for synaptic fitness. a** Western blot shows that PU-AD treatment rebalances the activity of synaptic protein pathways altered by tau overexpression in N2a cells. Graph, mean ± SEM, one-way ANOVA, Dunnett's post hoc, $n = 3–4$ independent experiments. **b** Schematic summary of synaptic pathways altered in both AD and N2a-tau cells through epichaperome network formation. Functional effectors of protein pathways chosen for validation in (**a**) are highlighted with a red outline. See also Fig. 5. **c** Western blot analyses show no HSF-1 activation under conditions where both tau and phosphorylated tau levels decrease upon PU-AD treatment. MDA-MB-468 cancer cell lysates were used as a control for the presence of active HSF-1 (indicated with an asterisk). Levels of the inducible heat shock proteins HSP70 and HSP27, surrogate markers of HSF-1 activation, were also probed. The expression level of individual chaperome members (HSP90s, HSC70, HSC60, HSP110) are also shown. Graph, mean ± SEM, unpaired two-tailed $t$-test, $n = 3$. **d, e** Viability of iPSC-derived neurons cultured for 24 h with Vehicle (V) or PU (0.5 μM PU-AD) (**d**) or for 72 h with concentrations of PU-AD from 0 to 100 μM (**e**). Graph, mean ± SEM, $n = 12$ for (**d**) and $n = 4$ for (**e**). *$P < 0.05$, **$P < 0.01$, ***$P < 0.001$. Source data are provided as a Source Data file.

Moreover, PU-AD delivery was associated with increased levels of the pro-survival x-linked inhibitor of apoptosis protein (XIAP), a protein proposed to prevent neuronal death in vivo[44], validating observations at the protein network level (Fig. 7a).

Cognitive decline in PS19 mice was measurable from 6 to 11 months of age using the Barnes maze (Supplementary Fig. 6A, B). To address the effect of PU-AD on memory and learning, and as a functional readout of PCBD, we designed a treatment

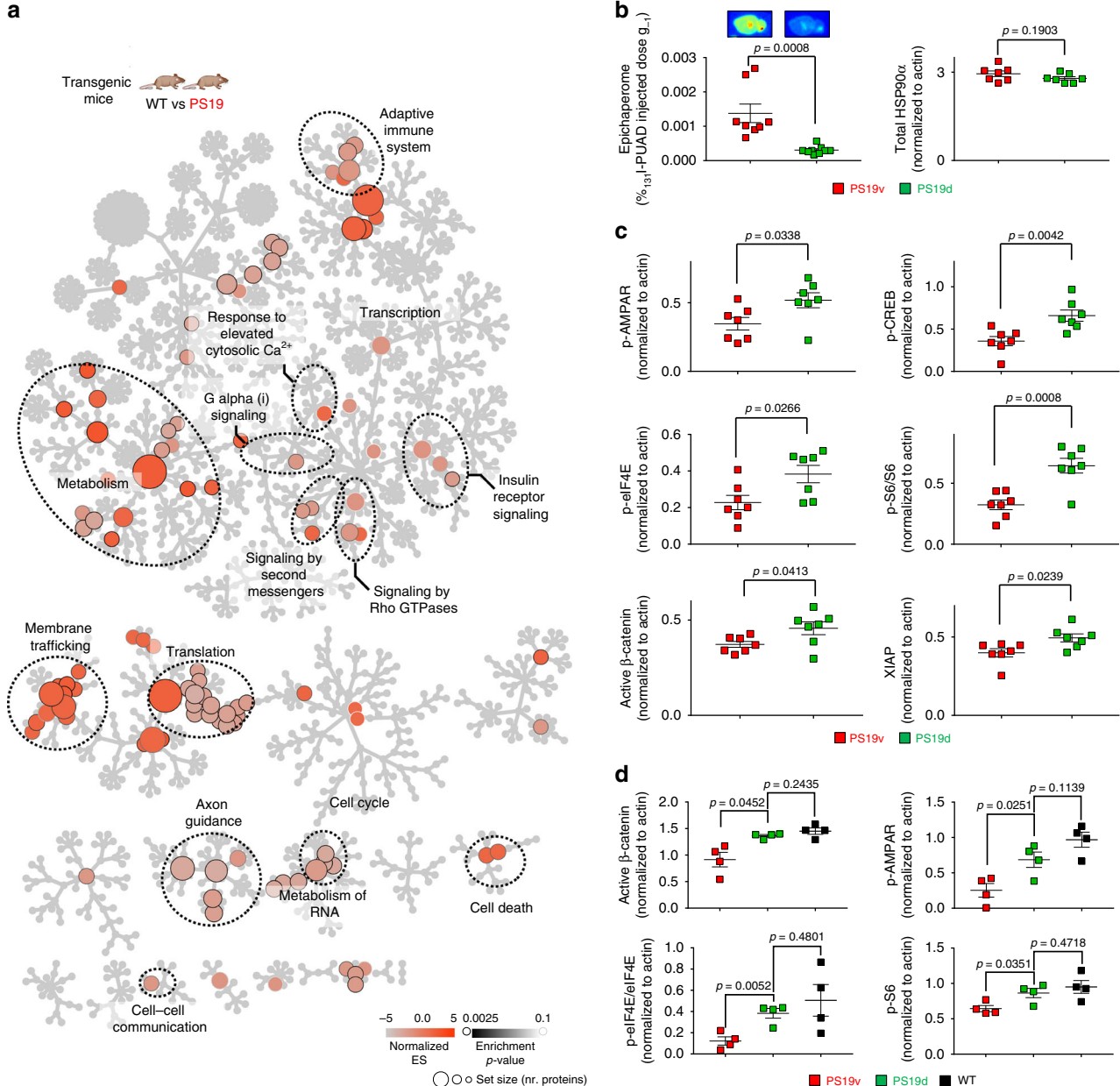

**Fig. 7 Epichaperome inhibition corrects the activity of synaptic protein networks in PS19 mice. a** Reactome pathway analysis shows the identity of epichaperome-mediated alterations in protein pathways intrinsic to PS19 mice. **b** Co-injection of PU-AD and tracer amount of $^{131}$I-PU-AD confirms target engagement in PS19 brains by PU-AD. Graph, mean ± SEM, unpaired two-tailed $t$-test, $n = 8$ for vehicle treated PS19 (PS19v) and $n = 9$ for PU-AD treated PS19 mice (PS19d). **c** Western blot of PS19 treated mice monitoring effects of protein pathways as in (**a**). We measured several effectors in the function of synaptic pathways: p-AMPAR Ser845, p-CREB Ser133 and active β-catenin. p-S6 and p-eIF4E are both key effectors linking insulin receptor signalling, via mTOR and other upstream signalling pathways, to translation initiation. XIAP is a potential inhibitor of neuronal apoptosis. Graph, mean ± SEM, unpaired two-tailed $t$-test, $n = 7$ PS19v, $n = 7$ PS19d. Mice were sacrificed 48 h post-last injected dose. See also Fig. 8a and Supplementary Fig. 5A. **d** A rebalance in the function of synaptic protein pathways to levels observed in WT mice was observed in PS19 mice treated with PU-AD. Graph, mean ± SEM, unpaired two-tailed $t$-test with Welch's correction, $n = 4$ PS19v, $n = 4$ PS19d and $n = 4$ WT. Mice were sacrificed 24 h post-last injected dose. See also Fig. 8a and Supplementary Fig. 5B. Source data are provided as a Source Data file.

paradigm in which PS19 mice received PU-AD (PS19d) or vehicle (PS19v) starting at 3 months of age until 7 months of age (Group 1). A second group, (Group 2), treatment started at 8–9 months of age and continued until 11–12 months of age (Fig. 8a). Based on a recent analysis of tau pathology progression in PS19 mice[30], these age points span pre-symptomatic to late-stage disease (Group 1, Braak Stage I/II to Stage III/IV and Group 2, Braak Stage III/IV to Stage V/VI, ref. [30]). PU-AD treatment significantly

improved performance in all measured criteria in both treatment paradigms (PS19v versus PS19d in the acquisition phase; Group 1: number of errors, $P < 0.0001$, $F (1, 110) = 19.77$; latency, $P < 0.0001$, $F (1, 110) = 47.76$; and % success, $P < 0.0001$, F $(1, 110) = 643.9$; two-way ANOVA and Group 2: number of errors, $P < 0.0001$, $F (1, 75) = 35.19$; latency, $P < 0.0001$, F $(1, 75) = 17.60$; and % success, $P < 0.0001$, F $(1, 75) = 19.91$; two-way ANOVA). PU-AD treated PS19 mice performed as well as the age-paired

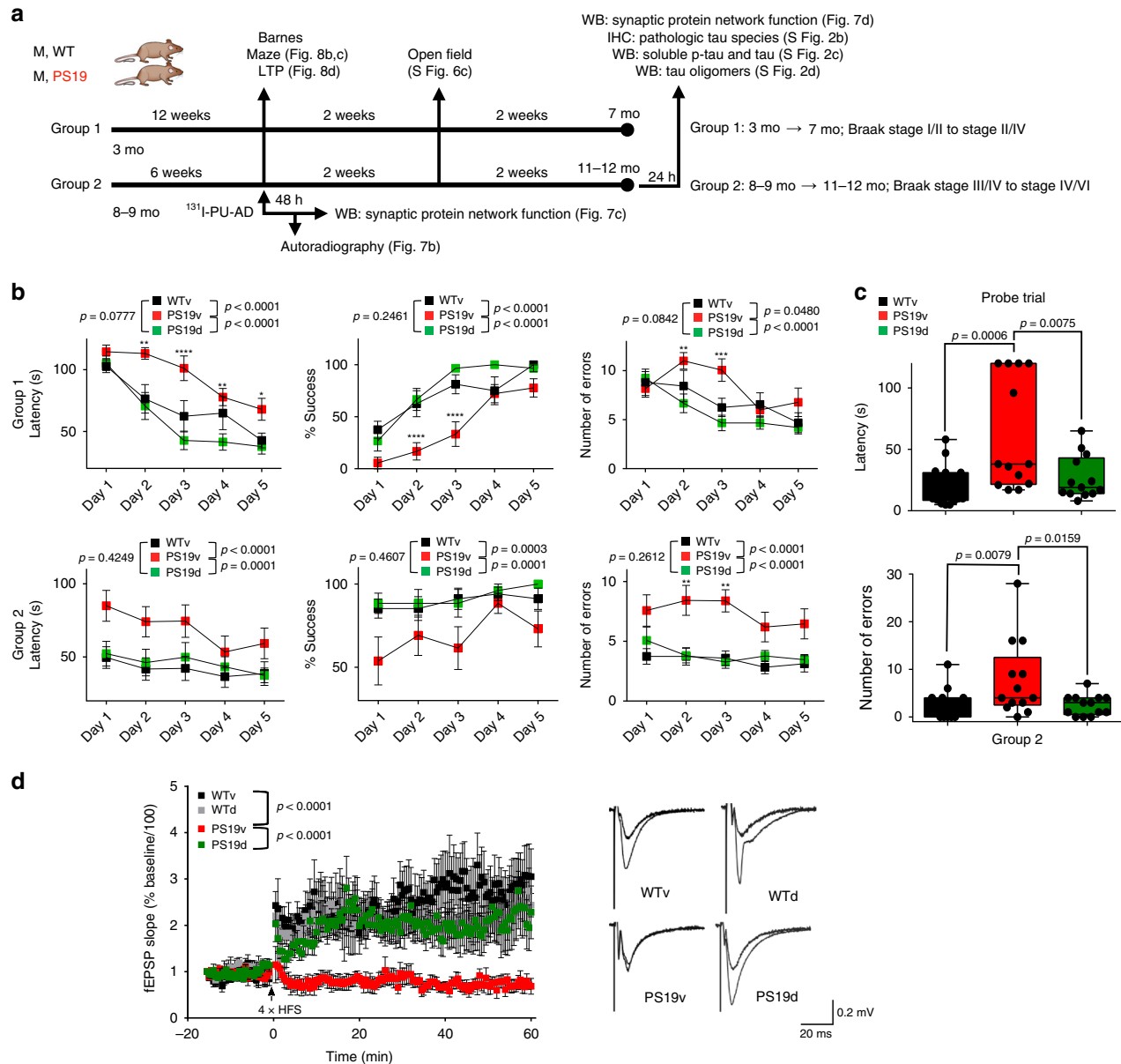

**Fig. 8 Epichaperome inhibition restores synaptic plasticity and spatial learning and memory in PS19 mice. a** Schematic of the treatment and testing paradigm. **b**, **c** Spatial learning and memory evaluated in the Barnes maze in PS19 treated with either PU-AD (PS19d) or vehicle (PS19v), and in vehicle treated WT mice (WTv) as in (**a**). Acquisition phase: two-way ANOVA, Bonferroni post hoc (**b**). Probe test (probe memory retention trial performed 24 h after the last acquisition session): one-way ANOVA, Dunnett's post hoc (**c**). Group 1; $n = 8$ WTv, $n = 9$ PS19v, $n = 15$ PS19d; Group 2; $n = 17$ WTv, $n = 13$ PS19v and $n = 13$ PS19d. See also Supplementary Fig. 6B. **d** Long-term potentiation (LTP) induced in slices from WT and PS19 mice treated with vehicle or PU-AD as indicated in (**a**), Group 2. Representative traces are also shown. Graph, mean ± SEM, two-way ANOVA, $n = 7$ WTv, $n = 6$ WTd, $n = 5$ PS19v and $n = 10$ PS19d. ****$P < 0.0001$, ***$P < 0.001$, **$P < 0.01$, *$P < 0.05$. Source data are provided as a Source Data file.

WT mice (Fig. 8b, c). In Group 2, which displayed the highest epichaperome levels, long-term memory impairments were found in PS19 mice (number of errors, $P = 0.0074$, $F (2, 40) = 5.56$; latency, $P = 0.0009$, $F (2, 40) = 8.432$). This was also significantly improved upon PU-AD treatment, where we observed treated PS19 mice to behave statistically the same as age-paired WT mice. PU-AD had no significant effect on locomotor activity (Supplementary Fig. 6C) and on spatial memory in WT mice (Supplementary Fig. 6D). We observed similar effects of PU-AD in 3xTg-AD mice, where similar to PS19 mice, epichaperome inhibition restored long-term memory in 3xTg-AD mice to levels in WT mice (Supplementary Fig. 7A–D) and significantly reduced pathologic human tau levels without affecting mouse tau species (Supplementary Fig. 7E).

**PCBD and compromised synaptic plasticity.** We next asked whether epichaperome inhibition leads to changes in synaptic transmission resulting from correction of PCBD and synaptic protein network rebalancing, possibly underlying observed spatial memory recovery. Spatial memory is encoded in the synapses of the trisynaptic circuit[45] that we find consists of the brain regions with the highest and earliest switch of the chaperome into epichaperomes, and in turn PCBD, namely the ENT, HIP and SUB, mnemonic-related temporal lobe regions which accrue neuropathology and synaptic dysfunction early and pervasively in AD[46,47]. We assessed long-term synaptic plasticity using slice electrophysiology in PS19 mice (treatment Group 2, Fig. 8d). Field excitatory postsynaptic potentials (fEPSCs) were recorded in the CA1 region of the hippocampus by stimulating the Schaffer

collaterals, and LTP was induced using a standard high-frequency stimulation (HFS) protocol[48]. We found LTP, thought to underlie learning and memory[48], was significantly impaired in PS19 mice compared to WT controls, as previously reported[26], and restored to WT level after PU-AD treatment ($P < 0.0001$, $F (3, 2744) = 303.7$; two-way ANOVA).

**Chronic epichaperome inhibition in mice.** To develop PU-AD for potential clinical use, we conducted a standard safety study of PU-AD administered chronically to healthy C57/BL6 mice (Supplementary Fig. 8A–C). We dosed mice three times per week for 90 to 180 days (40–70 PU-AD doses) with either vehicle control or PU-AD at a dose of 75 mg kg$^{-1}$ (i.e., target saturating dose under i.p. administration). No mortality or morbidity was observed during treatment or observation periods. All mice were sacrificed 24 h after final test article administration. Complete necropsies and analyses of haematology and clinical chemistry were conducted. No significant weight loss was found. Haematological and clinical chemistry findings were normal. Histopathology of major organs showed no toxic changes induced by PU-AD treatment. Having established that PU-AD can be administered chronically and is safe, we evaluated its effect when given over the entire adulthood of PS19 mice (>250 PU-AD doses; 3× per week 75 mg kg$^{-1}$ target saturating under i.p. administration) (Supplementary Fig. 8D). In PS19 mice, tau pathology typically results in decreased survival due to severe paralysis, however PU-AD administration from 3 months of age until death was well tolerated and significantly increased survival, with 50% of treated male mice having a 100–150-day delay in the terminal paralysis phenotype ($P = 0.0022$). Our findings were confirmed and substantiated under IND-enabling studies, and PU-AD has recently transitioned to clinical investigation in human patients[49].

## Discussion

We demonstrate cognitive deficits in AD stem from an inability to encode new information due to proteome-wide connectivity defects. These manifest in dysfunctions within protein networks and are executed through the switch of the chaperome into epichaperomes. It is important to note that such structural chaperome restructuring can lead to a loss of normal function as well as a gain of aberrant function in protein networks, with or without actually changing the expression level of an individual protein. We introduce herein the term PCBD to define this novel, clinically relevant mechanism. In this new framework, AD is a PCBDopathy, a disease of proteome-wide connectivity defects mediated by a maladaptive switch of the chaperome into epichaperomes. These proteome-wide connectivity disturbances are pharmacologically targetable by correcting PCBD through de-connecting aberrant epichaperome structures, and may be non-invasively detected in at-risk patients through molecular imaging.

Our data support a functional and causal link between global disturbances in dynamic protein networks in AD and the structural rewiring of the chaperome system into epichaperomes. These epichaperome structures provide the backbone upon which proteome-wide connectivity, and in turn, protein networks become disturbed and ultimately dysfunctional, supporting the concept that mechanistically AD is a PCBD. Among most sensitive to PCBD are pathways with key roles in synaptic plasticity.

We identify a common vulnerability of neuronal lineages to stress. Various individual stressors converge to create imbalances in a defined and specific set of neuronal protein pathways important for synapse formation and function, among other key pathways negatively impacted by AD. Our findings that stressors and vulnerabilities, including those associated with AD, rewire

proteome-wide connectivity, and thus cellular function, through maladaptive chaperome networks termed epichaperomes provide proof-of-principle for such a core, unifying, AD mechanism. By identifying an underlying shared trait in AD cases, our results point to a common AD mechanism, which we have demonstrated to be targetable and potentially druggable.

Our discoveries were enabled by a unique functional proteomics-bioinformatics pipeline referred to as chemical chaperomics which does not provide a simple list or inventory of the AD proteome. Rather chaperomics informs, in a large-scale unbiased manner, how global PPIs and the function of protein networks therein, change in response to AD-related factors. By capturing dynamically disease-altered protein connectivity networks with an innovative toolkit of chemical probe-based baits and then subjecting the cargo to unbiased MS followed by bioinformatics analyses, chaperomics is a quick, reproducible and versatile omics methodology that informs investigators into the functional significance of protein connectivity changes. These features provide chaperomics the power to investigate and uncover AD biology currently unavailable though classical omics-based methods in animal and cellular models as well as in live humans and postmortem tissues. Whereas individual genes and/or encoded proteins within neuronal pathways have been demonstrated to have defects either through genomic, transcriptomic, or shotgun-based proteomic studies[50–54], these methods, unlike chaperomics, remain intrinsically limited in deciphering the complexities of protein network dysfunctions.

Another decided advantage of chaperomics is that the datasets are enriched for functional protein pathways, unlike conventional transcriptomic approaches. Therefore, bioinformatics platforms are straightforward and easy to comprehend, rather than adding additional complexity to these large protein connectivity-based datasets.

Currently, the AD research community is facing many different challenges in elucidating the potential functions of gene variants and environmental stressors in the initiation and propagation of AD. Our data provide a confluence of evidence from cellular model, animal model, live human, and postmortem human brain tissues that chaperomics has the ability to discover the identity of proteome-wide alterations induced by individual stressors. Chaperomics therefore may enable many inquiries into AD biology, from screening for changes in protein function, PPIs, protein complexes; screening genotype-to-phenotype effects; discovery of new targets for AD treatment; and classification of AD based on functional proteome signatures, among others.

In the progression to AD, stressors and vulnerabilities such as aging, genetics, environmental factors and others, drive changes in the brain that occur and accumulate over decades. These result in imbalances in the connectivity of the neuronal circuitry, but also intracellularly, in the connectivity of neuronal proteins and protein pathways. An investigation through the use of chaperomics in a linear fashion: region, disease state, neuropathology, cognition, and sex into the dysregulation of PCBD from no cognitive impairment, to MCI to AD using postmortem tissues may therefore provide the causal and functional link between stressors and proteome dysfunction as it manifests during AD progression.

We also propose a novel modality for treatment by manipulation of specific neural populations affected by pathologic chaperome functions. Unlike individual molecular chaperones which may be protective by enabling protein folding[55], the formation of the epichaperome can inform either a loss of normal function or a toxic gain of function of the chaperome. Therefore, the chaperome is a regulator of normal protein networks, whereas the epichaperome, mediates and sustains aberrant protein networks in AD, which we believe leads to PCBD.

We show that functional imbalances in synaptic protein networks are reverted to normal upon pharmacologic epichaperome inhibition, demonstrating a causal link between dysfunctions in neuronal protein pathways and the epichaperome in AD, and highlighting the vital role of the epichaperome networks in molecular synaptic dysfunction. Epichaperome inhibition via PU-AD alone was sufficient to restore postsynaptic responsiveness and repair a largely compromised synaptic plasticity response. This treatment also attenuated PCBD as we observed reversal of cognitive decline, slowing of disease progression, and a delay in reaching the terminal disease stage, and significantly extending the lifespan of PS19 mice. Studies conducted in both preventive and interventional treatment settings demonstrated significant cognitive improvements without unwanted side effects.

Opportunities in selectively inhibiting the epichaperome over more abundant cellular pools of chaperome components, arise from the nature of the epichaperome itself, which presents as stable and multi-partner complexes formed by strong interactions, as opposed to other cellular forms of chaperones present in dynamic complexes of weak interactions and with a limited number of partners[17,18,20,24]. These structural and dynamic features provide an approach for the specific binding of small molecules (such as PU-H71 and PU-AD) through kinetic selectivity to differentiate the epichaperome from the more abundant chaperome[17,20]. This is a new avenue for rational therapeutics aimed at correcting PCBD through de-connecting aberrant epichaperome structures and the proteome-wide pathways they rewire. It will therefore be interesting, and certainly useful from a therapeutic perspective, to investigate whether modulation of other hubs in the epichaperome network (such as of HSC70 and other candidates) may induce, or not, epichaperome collapse, and in turn correct PCBD.

Radiolabeled PU-H71 and PU-AD are epichaperome detection reagents, akin to how an antibody can recognize a specific protein, as they dissociate from HSP90 residing in epichaperomes much more slowly (i.e., over several hours to days) than they do from other HSP90 pools (i.e., minutes to few hours); this difference in the $k_{off}$ (i.e., dissociation constant) provides epichaperome selectivity[17,20,25]. Importantly, these validated probes can be used to detect and quantify epichaperomes by positron emission tomography (PET) imaging in human patients[25].

An increasing number of genetic studies suggest that AD and cancer pathogenesis have commonalities[56,57]. Both diseases involve disruption of similar biological processes in the absence of a final common path, as the end result is very different, with early and pervasive neurodegeneration in AD and uncontrolled proliferation in cancer. We demonstrate that rewiring the chaperome machinery into epichaperome networks, which we recently identified to provide a survival advantage in cancer[17], is also at the core of pathological neuronal changes leading to synaptic dysfunction and subsequent memory impairment in AD.

In summary, our study fills a critical knowledge gap by revealing an underlying protein connectivity disturbance in AD, providing a novel mechanism how intra- and inter-neuronal network disturbances are mediated in this neurodegenerative disorder with profound public health implications. Although it is premature to suggest a precise sequence of PCBD spread throughout vulnerable brain circuits, we expect that early changes in PCBD will be seen in the hippocampus and continue to other hubs driving executive function (e.g., default network mode) as the disease progresses. We demonstrate that PCBD is a functional manifestation of the epichaperome and may be considered a biomarker of disease onset, as PCBD is not found to date in the brains of WT mice or age-matched cognitively normal subjects. In addition to a new AD mechanism and classification of AD as a PCBDopathy, we provide a viable therapeutic strategy to inhibit AD-specific pathologic protein networks, and a biomarker to detect such protein network vulnerability in living patients at risk for dementia. In sum, our study offers a novel modality for AD diagnosis and treatment that is based on protein connectivity dysfunction.

## Methods

**Reagents**. PU-AD, [131]I-PU-AD, [124]I-PU-AD, PU-beads (PU-bait) and control beads (CB), and PU-FITC were synthesized using previously reported methods[22,58].

**Mouse cohorts**. Two separate cohorts of transgenic mice were used in this study: (1) male and female hemizygous PS19 harbouring P301S in which the mouse prion promoter drives overexpression of the human T34 isoform of tau (1N4R) which carries the P301S mutation[26] and (2) male and female 3xTg-AD[59]. 3xTg-AD mice are widely used as a model of AD; they express human APP$_{swe}$, PS1$_{M146V}$, and tauP301L and progressively develop both Aβ and tau pathology in disease-relevant brain regions and in an age-dependent manner[59]. The PS19 strain breeds as hemizygotes and the non-transgene containing offspring were used as control animals for transgenic littermates. The 3xTg-AD model was generated using mice of a mixed background; C57BL/6 and 129S1/Sv. Some studies have used as control mice the original background strain provided by Dr. Frank LaFerla (C7BL/6;129 × 1/SvJ;129S1/Sv). In our study, a separately bred strain of WT mice, B6129SF2/J, of an identical mixed background were used to control for 3xTg-AD experiments[60]. PS19 mice were bred and housed with same-sex littermates at Memorial Sloan Kettering Cancer Center (MSKCC). 3xTg-AD mice were purchased from The Jackson Laboratories and housed at MSKCC. All procedures were approved by the MSKCC Institutional Animal Care and Use Committee. All animal studies were conducted in compliance with MSKCC's Institutional Animal Care and Use Committee guidelines.

**Human brains**. All human brain tissue samples were obtained with appropriate consent given before collection of tissue. Tissue samples were obtained from the following tissue banks regulated and approved by the NIH Neurobiobank (NBB): New York Brain Bank at Columbia University, University of Maryland Brain, and Tissue Bank and Human Brain and Spinal Fluid Resource Center, Los Angeles. Postmortem intervals were <24 h. Age-matched normal brain tissues were from human subjects with no evidence of clinical cognitive impairment or neurological disorders.

**Cell culture and plasmid transfection**. Mouse neuroblastoma N2a cells were obtained from American Type Culture Collection (ATCC) and were grown in Dulbecco's modified Eagle's medium (DMEM), supplemented with 10% foetal calf serum, 100 U/ml penicillin and 100 μg/ml streptomycin at 37 °C in 95% humidified air with 5% $CO_2$. pCMV6-Tau (4R0N) and eYFP-tagged pcDNA3-Tau (4R2N; T40) were transfected into N2a cells using Lipofectamine 2000 (Life Technologies, Bethesda, MD) according to the manufacturer's protocol. Opti-MEM I Reduced Serum Medium was employed for transfection. After six hours of transfection, transfection media was replaced with 10% FBS containing fresh media. For PU-AD treatment, cells were treated for 8 h after 36 h of plasmid transfection, and then harvested. Cells were authenticated using short tandem repeat profiling and routinely tested for mycoplasma.

**Drug treatment of human AD iPSC-derived neurons**. Existing and well-described human iPSC lines were used in this study, representing healthy controls (CV4a, NDC5; described in Israel et al.[61] and van der Kant et al.[62]), familial AD caused by a duplication of the APP gene on chromosome 21 (APPdp; described in Israel et al.[61]) or APP KM670/671NL (Swedish mutation; APPswe 2.1, 1.2 lines; described in Woodruff et al.[63]), or sporadic AD (SAD2, SAD7; described in van der Kant et al.[62]). Neural progenitor cells (NPCs) lines were differentiated on PA6 stromal cells and purified from as described previously[61–63]. NPCs were seeded onto 10 cm plates and grown to 60% confluence in NPC base medium (DMEM/F12, B-27, N-2, Pen/Strep) containing FGF, then differentiated into forebrain-like neurons by FGF withdrawal for 21 days[61–63]. This method has been shown to produce cultures of 80–90% neurons, with 10–20% of the remaining cells being non-neuronal cells that include glia and undifferentiated neural progenitors, and because these are not purified neurons, we refer to these mixed-cell populations as differentiated NPCs (dNPCs). After 21 days differentiation by FGF withdrawal, dNPCs were dissociated using Accutase and manual trituration, counted, and re-plated onto 12-well plates (10^6 live dNPCs/well) and allowed to recover for 14 days prior to drug treatment. PU-AD was dissolved in DMSO and subsequently diluted into NPCb at a final concentration of 0.1% DMSO (vehicle = 0.1% DMSO in NPCb). dNPCs were treated with vehicle or 10 nM, 100 nM, or 500 nM PU-AD for 24 h then washed, pelleted, and frozen at −80 °C until analysis. For viability experiments, NPCs were plated, differentiated, dNPCs were replated, and cells were treated for 24 h with vehicle or 500 nM PU-AD drug in vehicle, as described above. During the final 2 h of treatment, 1/10th volume of 10x resazurin

solution (500 μM, dissolved in vehicle) was added to the existing medium, cells were returned to the incubator for 2 h, then fluorescence of resorufin (reduced form of resazurin) was measured using a Molecular Devices SpectraMax i3 fluorescence plate reader (ex 548 nm/em 588 nm). Background-subtracted fluorescence values were normalized to fold-change versus vehicle for each cell line (where vehicle = 1.0). All work with human iPSC-derived cell lines was approved by the Humboldt State University and UC San Diego Institutional Review Boards and ESCRO committee in accordance with NIH guidelines.

**Long-term neuronal toxicity study.** Human embryonic stem cells (hESCs) were differentiated with a modified dual-SMAD inhibition protocol towards floor plate-based midbrain dopaminergic (mDA) neurons as described previously[64]. Briefly, hESCs were maintained on mouse embryonic fibroblasts and passaged with Dispase (STEMCELL Technologies). For each differentiation, hESCs were harvested with Accutase (Innovative Cell Technology). At day 30 of differentiation, hESC-derived mDA neurons were replated and maintained on dishes precoated with polyornithine (PO; 15 μg mL$^{-1}$), Laminin (1 μg mL$^{-1}$), and Fibronectin (2 μg mL$^{-1}$) in Neurobasal/B27/L-glutamine-containing medium (NB/B27; Life Technologies) supplemented with 10 μM Y-27632 (until day 32) and with brain-derived neurotrophic factor (BDNF; 20 ng mL$^{-1}$; R&D), ascorbic acid (AA; 0.2 mM, Sigma), glial cell line-derived neurotrophic factor (GDNF; 20 ng mL$^{-1}$; R&D), transforming growth factor type β3 (TGFβ3; 1 ng mL$^{-1}$; R&D), dibutyryl cAMP (0.5 mM; Sigma), and DAPT (10 nM; Tocris). Two days after replating, mDA neurons were treated with 1 μg mL$^{-1}$ mitomycin C (Tocris) for 1 h to kill any remaining non-post mitotic contaminants. Toxicity assays were performed at 65 days of neuron differentiation. PU-AD was added to the cells for 72 h and the CellTiter-Glo Luminescent Cell Viability Assay (Promega) assay was performed according to the manufacturer's indications.

**Behavioural studies.** Male mice were tested for spatial memory and locomotion using Barnes maze (San Diego Instruments; model #7001–0235) and Openfield (Actitrack, Panlab, Barcelona, Spain), respectively. All the videos were recorded and analysed for distance (m) and speed (ms$^{-1}$) using the ANY-maze Video Tracking System (v4.94 h beta, 4.112, Stoelting). Animals were placed in the behavioural room for habituation under low-intensity light conditions for 1 h before each experiment. This non-stressful light condition was retained during the experiments, unless otherwise specified. Mice were assigned by chance, rather than by choice, to either the PU-AD or vehicle group. No statistical methods were used to pre-determine sample sizes but these are similar to those generally employed in the field.

**Barnes maze.** Mice were tested for hippocampal-dependent spatial memory on the Barnes maze (San Diego Instruments). The basic function of Barnes maze is to measure the ability of a mouse to learn and remember the location of an escape hole using distal visual cues located around the testing area, and is based on rodents' aversion of open spaces, which motivates the test subject to seek shelter in the escape box. The protocol was described previously by the Lee laboratory[27]. Briefly, mice were allowed to habituate to the environment for 1 h before testing and to the maze for 30 s before each trial. The light conditions were adjusted from dim intensity during the habituation period to a higher intensity during the trial sessions to promote exploration. The maze was cleaned with 70% ethanol between animals to minimize odour cues. The experiment was performed on 6 sessions conducted on 6 consecutive days. During the first five sessions, mice were exposed to the maze for three 2.5 min trials that were 15 min apart. The first trial was not scored and mice that did not find the escape compartment were guided there and allowed to remain for 30 s. Mice were then tested for their ability to remember the fixed position of this escape compartment during trials two and three; this paradigm tested the short-term retention memory of mice. One day after the acquisition phase, thus on day 6 (probe memory retention trial), the animal was tested in a single trial; this paradigm was designed to test the long-term retention memory of mice. The performance was scored using different parameters: latency, success rate and number of errors. The latency (in seconds) was measured as the time spent in the platform to find the escape hole and the number of errors performed during this time. The success rate was estimated per individual based on the ability to find the escape in <2 min. All the parameters per session were compared among groups. Statistical analyses were performed using Prism 6 GraphPad software.

**Open field.** The open field test is a common measure of exploratory behaviour and general activity in both mice and rats, where both the quality and quantity of the activity can be measured. Principally, the open field (OF) is an enclosure, square, in shape with surrounding PVC walls that prevent escape. The most basic and common outcome of interest is "movement". This allows for the analysis of the level of activity, the zonal distribution and the movement (anxiety and exploratory index). The Panlab Infrared (IR) Actimeter (Actitrack, Panlab, Barcelona, Spain) allows the study of spontaneous locomotor activity and exploration in rodents. The system is composed by a two dimensional ($x$ and $y$ axes) square frame, a frame support and a control unit. The infrared frame consists of 25 × 25 cm frame containing 16 × 16 infrared beams at 1.5 cm intervals for optimal subject detection. The infrared photocell system was set with up to 4 sensitivity level. The frames

were registered by the LE8825 unit. The system is completely modular: each frame may be used for evaluation of general activity, locomotor and stereotyped movements or rearings. Mice were placed in the centre of the open field arena and after 2 min the activity was recorded for 4 min. In this period, the animal's movement was measured using the ANY Maze automated tracking system. 70% ethanol was used to clean the surface between animals and sessions.

**Electrophysiological analysis.** Eleven mo WT and PS19 mutant mice were deeply anesthetized with isoflurane and then decapitated. Brains were rapidly removed and immersed in ice-cold oxygenated (95% O$_2$ and 5% CO$_2$) dissection buffer containing (in mM): 83 NaCl, 2.5 KCl, 1 NaH$_2$PO$_4$, 26.2 NaHCO$_3$, 22 glucose, 72 sucrose, 0.5 CaCl$_2$, and 3.3 MgCl$_2$. After brain hemispheres were separated, coronal slices (400 μm) were cut using a vibratome (VT1200S, Leica), incubated in dissection buffer for 40 min at 34 °C, and then stored at room temperature for reminder of the recording day. All slice recordings were performed at 34 °C unless otherwise specified. Slices were visualized using IR differential interference microscopy (DIC) (BX51, Olympus) and a CMOS camera (Orca-Flash4.0LT, Hamamatsu). Placements of electrodes were visualized with a 10× (0.3 NA) or a 60× Olympus water immersion (1.0 NA) objective. For all experiments, external recording buffer was oxygenated (95% O$_2$ and 5% CO$_2$) and contained (in mM): 125 NaCl, 25 NaHCO$_3$, 1.25 NaH$_2$PO$_4$, 3 KCl, 25 dextrose, 1 MgCl$_2$, and 2 CaCl$_2$. Recording microelectrodes were fabricated from borosilicate glass (Sutter Instrument) to a measured tip resistance of 1–3 MΩ, and were filled with external recording buffer. Synaptic responses were recorded in the stratum radiatum of CA1 by stimulating Schaffer collateral/commissural pathway with a tungsten bipolar electrode (Harvard Apparatus). Input/output curves were first generated using 0.1 ms pulses with stimulating intensities from 0 to 520 μA at 40 μA increments. Pairs of fEPSPs were evoked with an interstimulus interval of 50 ms, and at ~50% maximum fEPSP amplitude. Baseline was established by recording synaptic responses every 30 s for 15 min, followed by LTP induction using HFS consisting of four 100 Hz trains at 20 sec intervals. fEPSP were recorded every 30 s for 60 min after HFS. Signals were amplified with a Multiclamp 700A amplifier (Molecular Devices), digitized with an ITC-18 digitizer (HEKA Instruments Inc.) and filtered at 2 KHz. Data were monitored and acquired using Axograph X software. Data analysis was performed using Axograph X built-in analysis and IGOR Pro software (Wavemetrics) on a Macintosh computer.

**Epichaperome determination in mice.** Epichaperome detection in live mice was performed using a labelled epichaperome probe, as reported[17]. Specifically, 35 μCi g$^{-1}$ of $^{131}$I-PU-AD (7 μCi μL$^{-1}$ in saline, Hospira, 0.9% sodium chloride in water) was injected intravenously into mice. Both female and male mice were used. After 24 h, and prior to brain excision, mice were perfused with 20 mL PBS. Brains were cut in half along the superior sagittal sinus. Hemispheres were embedded in agarose gel (5% in water). Serial sagittal and coronal sections of 80 μm in thickness were cut on a Leica VT 1000 S vibratome and were then mounted on microscope slides. Digital autoradiography was performed by placing tissue sections in a film cassette against a phosphor imaging plate (Fujifilm BAS-MS2325; Fuji Photo Film) for a 4-day exposure period at −20 °C. Phosphor imaging plates were read at a pixel resolution of 25 μm with a Typhoon 7000 IP plate reader (GE Healthcare). After auto-radiographic exposure, the same frozen sections were then dried and darkfield images were taken on an Observer Z1 microscope (Carl Zeiss, Germany) with 5 × / 0.15NA objective and ZEN2.3 acquisition software. For epichaperome analysis and quantification, internal standards were included in each autoradiography cassette and image analysis was performed using ImageJ 1.48v. Regions of interest were digitally drawn around respective brain regions and background corrected values were converted into relative activities using the internal standard. Images for anatomical location of brain areas were obtained from the Allen Institute for Brain Science. Allen Brain Atlas API. Available from: brain-map.org/api/index.html, Mouse, P56[65].

**Epichaperome engagement in cell lines.** For Fig. 2f, N2a cells were transfected with an eYFP-tau containing plasmid (a kind gift from Dr. Marc Diamond) using the Lipofectamine method (Invitrogen). Single colonies were obtained and selected using media supplemented with G418 (Gibco) and expanded for screening. Stably transfected clones were treated with indicated inhibitors for 24 h, then drug was washed off and new media was added for an additional 24 h. For Fig. 5d, N2a cells with transient overexpression of CMV6-Tau (4R0N) were used. Cells were lysed in 20 mM Tris pH 7.4, 20 mM KCl, 5 mM MgCl$_2$, 0.01% NP40 buffer containing protease and phosphatase inhibitors. For native gels, the protein extracts were loaded onto 4–10% native gradient gel and resolved at 4 °C. Gels were immunoblotted following a transfer in 0.1% SDS-containing transfer buffer for 1 h. Antibodies used were: HSP110 (SPC-195) (Stressmarq); HSC70 (SPA-815) and HOP (SRA-1500) (Enzo); HSP90α (ab2928) Abcam; and CDC37 (4793) (Cell Signalling Technology).

**Epichaperome analysis.** Analyses of the epichaperome were performed as previously reported[17]. In brief, samples were homogenized in either Bicine/Chaps buffer (20 mM bicine, pH 7.6, 0.6% CHAPS, ProteinSimple) or 20 mM Tris pH 7.4, 20 mM KCl, 5 mM MgCl$_2$, 0.01% NP40 buffer (+/− 0.6% CHAPS) containing

protease and phosphatase inhibitors. 300 µg of total protein was incubated with either PU-beads or control beads (10 µL) for 30 min. Supernatants were collected and run onto native gels. Hindred and eighty micrograms of protein was loaded onto a 4–10% native gradient gel and resolved at 4 °C. Gels were immunoblotted following a transfer in 0.1% SDS-containing transfer buffer for 1 h. Membranes were blotted with the HSP90α antibody from Abcam (ab2928) followed by anti-rabbit HRP-conjugated secondary antibody (Santa Cruz Biotechnology). Beads were washed five times with 20 mM Hepes pH 7.3, 50 mM KCl, 5 mM $MgCl_2$, 20 mM $Na_2MoO_4$, and 0.01% NP40 buffer and run on 7% SDS-PAGE. Membranes were blotted with the following antibodies: HSP90β (SMC-107) and HSP110 (SPC-195) (Stressmarq); HSC70 (SPA-815), HSP90β (SPA-845) and HOP (SRA-1500) (Enzo); HSP90α (ab2928) and AHA1 (ab56721) (Abcam); CDC37 (4793) (Cell Signalling). Blots were washed with TBS/0.1% Tween 20 and incubated with appropriate HRP-conjugated secondary antibodies. Chemiluminescent signals were detected with an Enhanced Chemiluminescence Detection System (GE Healthcare) following manufacturer's instructions.

**Fluorescence polarization assay.** For epichaperome inhibition measurements, assays were carried out in black 96-well microplates (Greiner Microlon Fluotrac 200). N2a cells transfected with pCMV6-Tau plasmid were treated with PU-AD for 24 h followed by 24 h drug wash off. Equal amounts of control and treated cell lysates (0.5 µg) were added to each well followed by PU-FITC (10 nM) in a final volume of 100 µl assay buffer (20 mM Hepes (K), pH 7.3, 50 mM KCl, 2 mM DTT, 5 mM $MgCl_2$, 20 mM $Na_2MoO_4$, and 0.01% NP40 with 0.1 mg ml$^{-1}$ BGG). For competition experiments conducted with AD brain homogenates, PU-AD was serially diluted in assay buffer in 96-well plates. In a separate tube, AD brain lysates diluted in assay buffer were mixed with PU-FITC (10 nM). To account for background signal, buffer and PU-FITC only controls were included in each assay. FP values in mP were measured every 5–10 min. The assay window was calculated as the difference between the FP values recorded for the bound fluorescent tracer and the FP value recorded for the free fluorescent tracer (defined as mP − mPf). Measurements were performed on a Molecular Devices SpectraMax Paradigm instrument (Molecular Devices, Sunnyvale, CA), and data were imported into SoftMaxPro6 and analysed in GraphPad Prism 7.

**Epichaperome detection by live imaging.** The microdose $^{124}$I-PU-AD PET-CT (Dunphy, M. PET Imaging of Subjects Using $^{124}$I-PU-AD available from: http://clinicaltrials. gov; NCT03371420) was approved by the institutional review board, and conducted under an exploratory investigational new drug (IND) application approved by the US Food and Drug Administration. Patients provided signed informed consent before participation. $^{124}$I-PU-AD tracer was synthesized in-house by the institutional cyclotron core facility at high specific activity. Analyses of the epichaperome by positron emission tomography (PET-CT) were performed as previously reported[17]. In brief, research PET-CT was performed using an integrated PET-CT scanner (Discovery DSTE, GE Healthcare). CT scans for attenuation correction and anatomic coregistration were performed before tracer injection. Patients received 185 megabecquerel (MBq) of $^{124}$I-PU-AD by peripheral vein over two minutes. PET data were reconstructed using a standard ordered subset expected maximization iterative algorithm. Emission data were corrected for scatter, attenuation, and decay. $^{124}$I-PU-AD scans (PU-AD PET) were performed 3 h after tracer administration. Numbers in the scale bar indicate upper and lower standardized uptake value (SUV) thresholds that define pixel intensity on PET images.

**Tissue preparation.** Mouse brains were hemisected and one hemisphere was fixed in 4% paraformaldehyde for immunohistochemistry and the other hemisphere was flash-frozen in liquid nitrogen. Regional dissections were performed, and select tissues homogenized in a buffer containing 50 mM Tris base, pH 8.0, 130 mM NaCl, 1 mM EDTA, protease inhibitor cocktail III (EMD Millipore, Darmstadt, Germany), phosphatase inhibitor cocktail(s) I and II (Sigma Aldrich, St. Louis, MO, USA), and 1 mM PMSF. The homogenate was spun for 10 min at 13,000 × g. The resulting supernatant was used as a total fraction for western blotting (fibrillary tau tangles, cell membranes, and other cellular debris were removed in the pellet). In other experiments, proteins were extracted from mouse brain tissues with 2% SDS. 10–100 µg of the protein extract was subjected to SDS-PAGE followed by western blotting procedure.

**Western blots.** Human brain samples were lysed in 20 mM Tris pH 7.4, 20 mM KCl, 5 mM $MgCl_2$, 0.01% NP40, and centrifuged at 13,000 × g for 15 min at 4 °C. For mice, dissected hippocampi were homogenized in radioimmunoprecipitation assay buffer (TBS with 1% NP-40, 1% sodium deoxycholic acid, 0.1% sodium dodecyl sulphate, and protease/phosphatase inhibitors) followed by centrifugation at 20,000 × g for 15 min at 4 °C. To measure soluble tau species, hippocampal, frontal cortex and cerebellum mouse brain samples were homogenized in 50 mM Tris pH 8.0, 130 mM NaCl, 5 mM KCl buffer with protease inhibitors added. To extract insoluble proteins, supernatants were removed, pellets were resuspended in 70% formic acid (FA) and centrifuged at 20,000 × g for 15 min at 4 °C. FA extracts were then neutralized in 1 M Tris base at the ratio of 1:20. Protein concentration of all the samples was determined by BCA method. Proteins were separated on

acrylamide gels by SDS-PAGE, and then transferred to PVDF or nitrocellulose membranes, blocked for 1 h in 5% milk in TBS, and incubated overnight with indicated primary antibody. The membranes were blotted with anti-human tau HT7 Ab (MN1000), anti-phospho-tau (Thr231) (MN1040), anti-phospho-tau (S202/T205) (MN1020/AT8) (ThermoFisher Scientific); APP (RU-369) (Rockefeller University); anti-phospho-tau (Ser262) (54973, AnaSpec), anti-HSP70 (SPA-810), anti-HSP60 (SPA-806), anti-HSC70 (SPA-815), anti-HOP (SRA-1500) (Enzo); anti-HSP27 (SPA-800) (Enzo); anti-HSP90α (ab2928), anti-phospho-HSF1 (Ser326) (ab76076) (Abcam); anti-HSP90β (SMC-107), anti-HSP/HSC70 (SMC-106), anti-HSP110 (SPC-195) (Stressmarq), anti-phospho-CREB1 (Ser133) (SC-81486) (Santa Cruz Biotechnology); anti-CREB (9197), anti-phospho-CREB1 (Ser133) (9196), anti-ERK (4695), anti-HSF1 (4356), anti-AMPA (13185), anti-phospho-AMPA (Ser845) (8084), anti-cofilin (5175), anti-phospho-cofilin (Ser3) (3313), CDC37 (4793), anti-eIF2α (5324), anti-phospho-eIF2α (Ser51) (3398), anti-phospho-eIF4E (Ser209) (9741), anti-eIF4E (9742), anti-β-catenin (9582), anti-active β-catenin (19807), anti-phospho-S6 ribosomal protein (Ser235/236) (4858), anti-S6 ribosomal protein (2217) (Cell Signalling Technology) and β-actin (A1978, Sigma-Aldrich). Blots were washed with TBS/0.1% Tween 20 and incubated with appropriate HRP-conjugated secondary antibodies (Southern Biotech, Birmingham, AL, USA). Chemiluminescent signals were visualized with Enhanced Chemiluminescence System (GE Healthcare) following manufacturer's instructions and quantified using image Studio Lite Ver. 5.2 (LI-COR Biosciences). Uncropped and unprocessed scans of the most important blots are provided in the Source Data file.

**Immunohistochemistry.** Brains were processed for paraffin embedding and serial coronal sections (5 µm thick) were obtained with the microtome (Leica RM2265) and mounted on Superfrost plus slides (ThermoFisher). Each slide contained sections of four groups, 2–3 sections per group, to ensure consistent staining conditions. IHC experiments were carried manually as follows: antigen retrieval was performed in 10 mM citric buffer, pH 6.0 in the microwave at 98 °C for 15 min. Sections were then treated for quenching endogenous peroxidase by immersing slides in 3% $H_2O_2$ in PBS—15 min at RT[66]. Subsequently, the sections were rinsed in PBS and treated with background blocking agent (INNOVEX Biosciences, Richmond, CA, USA) for 15 min. Slides were immersed in 0.3% Triton in PBS for 15 min. Antibodies were mouse monoclonal antibodies: anti-phospho Thr231 (1:100, AT180, ThermoFisher, Waltham, MA), anti-phospho Ser396/404 (PHF1, 1:100, gift of Dr. Peter Davies, Feinstein Institutes for Medical Research, Northwell Health) and anti-phosphoSer202 (CP13, 1:50, gift of Dr. Peter Davies, Feinstein Institutes for Medical Research, Northwell Health). Antibodies were prepared in PBS with 0.1% BSA and incubations performed overnight at 4 °C. Slides containing tissue sections were washed in PBS with 0.1% BSA and incubated in Max vision 1 and 2 (Maxvision Biosciences Inc, Bothell, WA), 30 min in each, at room temperature. Slides were developed for 1–3 min in diaminobenzidine solution (Sigma Aldrich, Saint Louis, MO) with 0.05% $H_2O_2$. Tissue sections were counterstained with hematoxylin. Detection of anti-human tau (HT7, 1:50, ThermoFisher, Waltham, MA) and anti-phospho Ser202/Thr205 (1:200, AT8, ThermoFisher, Waltham, MA) was performed using the Discovery XT processor (Ventana Medical Systems). Slides were blocked with the background blocking agent (INNOVEX Biosciences, Richmond, CA, USA) for 32 min. Primary antibody incubation was done for 3 h, followed by Maxvision-1 for 32 min and Maxvision-2 for 16 min. Detection was performed with the DABMAP kit (Ventana Medical Systems).

**Image analysis.** After dehydration and mounting in Permount (Fisher Scientific, Fairlawn, NJ), slides were scanned using Panoramic 250 digital slide scanner (3DHistech, Ltd, Budapest, Hungary) with 20x lens (NA 0.8). Telencephalic areas examined were identified on the basis of the anteroposterior and mediolateral coordinates of Paxinos and Franklin[67]. Regional structures were confirmed by the distinctive cytoarchitectonic features of the areas in haematoxylin and eosin stained sections. We used Metamorph Software (Molecular Devices, PA) to quantify the immunoreactive elements in 8–12 randomly selected sections from the anteroposterior regions. Each slide included 2–3 sections of each group (WT and PS19, treated and control groups). The first slide was separated by 300 microns from the following slide. Regions of study were designated by using Panoramic Viewer Software (3DHistech Ltd, Budapest, Hungary). Regions were exported into tiled tiff images and used for image analysis. The threshold for positive signal was established for each marker and used identically in all the groups. No difference in total area was identified due to shrinkage of tissue and the intensity was plotted as the average from the total area. The signal was expressed in percent of positive versus total area for each region and case (expressed as mean and standard error of the mean).

**Pharmacokinetic studies.** Pharmacokinetic studies were performed in male B6D2F1 mice (Harlan Laboratories) following intravenous (iv) administration of PU-AD. Blood and brain tissue were collected following euthanasia at the scheduled time points of 5, 30 min, 1, 4, and 8 h following iv dosing. Brain samples were immediately flash frozen in liquid nitrogen following collection and stored at −80 °C until analysis. Blood samples were collected into an EDTA tube by heart puncture and centrifuged at 13,500 rpm for 5 min at 4 °C to separate the plasma

and immediately frozen on dry ice and stored at −80 °C until analysis. The plasma sample (50 μL) was added with 250 μL of acetonitrile and 10 μL of internal standard (PU-AD-$d_4$), and then thoroughly vortexed to precipitate plasma protein and centrifuged at 13,500 rpm for 5 min at 4 °C. The supernatant was evaporated on a GeneVac and the residue was taken up into 100 μL of mobile phase (65% $H_2O$: 35% acetonitrile + 0.1% FA), applied to a 96-well plate and analyzed by LC-MS/MS (6410, Agilent Technologies) to measure concentrations of PU-AD. A Zorbax Eclipse XDB-C18 column (4.6 × 50 mm, 5 μm) was used for the LC separation, and the analyte was eluted under an isocratic condition (65% $H_2O$ + 0.1% HCOOH: 35% $CH_3CN$) for 5 min at a flow rate of 0.4 mL/min. Concentration in plasma was determined by comparison to a standard curve prepared by spiking untreated plasma samples to appropriate concentration of PU-AD (1–2000 ng/mL), processed and analysed in the same manner as treated samples specified above. Frozen brain tissue was dried, weighed and added with 750 μL of water/acetonitrile (7:3) and 10 μL of internal standard (PU-AD-$d_4$) solution. This was homogenized, extracted with methylene chloride and evaporated on a GeneVac. The residue was taken up into 100 μL of mobile phase (65% $H_2O$: 35% acetonitrile + 0.1% FA) and analyzed by LC-MS/MS (6410, Agilent Technologies) to measure concentrations of PU-AD. A Zorbax Eclipse XDB-C18 column (4.6 × 50 mm, 5 μm) was used for the LC separation, and the analyte was eluted under an isocratic condition (65% $H_2O$ + 0.1% HCOOH: 35% $CH_3CN$) for 5 min at a flow rate of 0.4 mL min$^{-1}$. Concentration in brain was determined by comparison to a standard curve of PU-AD (1–2000 ng/mL). Drug concentration (μM) in the brain was then modified to account for an average water space of 0.8 mL g$^{-1}$.

**Toxicology study**. This study assessed the safety and relevant toxicities of PU-AD administered by i.p. injection three times a week over a chronic administration period (3+ months). Healthy male and female C57/BL6 mice were used for this study. Seven mice (three male and four female) were administered the test article three times per week. Six mice (three male and three female) were administered the vehicle control. All animals were observed daily for mortality from the time of animal receipt through the end of the study. Body weights for all animals were recorded no more than three times, but no fewer than once, per week during the administration of the test article. All mice were observed for clinical symptoms at the time the animals were received and on all days in which the test article was administered. On the final study day, mice were anesthetized with isoflurane and ~50 μL of whole blood was collected from the orbital plexus of each mouse into a labelled tube containing EDTA anticoagulant. Within 2 h of blood collection, blood samples were analysed. A necropsy was performed on each animal. Tissues were collected into formalin. At the time of necropsy, gross examinations of each animal including internal organs were performed by a pathologist and any macroscopic lesions or other abnormal findings were recorded using standard terminology. For histopathology, all tissues were preserved in formalin. After >24 h in fixative, tissues were processed, embedded in paraffin, sectioned at 4 μm and stained with haematoxylin and eosin. All tissues were examined by a pathologist. Lesions were recorded using morphologic diagnoses following standardized nomenclature.

**Sample preparation for chaperomics**. For PU-bait affinity purification, protein extracts were prepared in 20 mM Tris pH 7.4, 20 mM KCl, 5 mM $MgCl_2$, 0.01% NP40 buffer with protease and phosphatase inhibitors added (Roche). Samples were incubated with the PU bait or the Control bait for 3 h at 4 °C, washed with 20 mM Tris pH 7.4, 20 mM KCl, 5 mM $MgCl_2$, 0.01% NP40 buffer and subjected to SDS-PAGE. The Control bait contained an inert molecule[17,22]. The samples were applied onto SDS-PAGE. The gels were stained with SimplyBlue Coomassie stain (ThermoFisher Scientific) and submitted for analysis. Gel lanes were cut into an average of 12 gel bands. Gel bands were completely destained with 50% methanol and 25 mM $NH_4HCO_3$/50% acetonitrile and diced into small pieces and dehydrated with acetonitrile and dried by vacuum centrifugation. The gel pieces were rehydrated with 10 ng μL$^{-1}$ trypsin solution (Trypsin Gold, Mass Spectrometry Grade, Promega) in 50 mM $NH_4HCO_3$ and incubated at 37 °C overnight. Peptides were extracted twice with 5% formic acid/50% acetonitrile followed by final extraction with acetonitrile. Samples were concentrated to a very small volume by vacuum centrifugation and peptides reconstituted in 2% acetonitrile/4% FA.

**MS data acquisition**. LC-MS/MS analysis was performed using a Q Exactive mass spectrometer coupled to a Thermo Scientific EASY-nLC 1000 (Thermo Fisher Scientific, Waltham, MA) equipped with a self-packed 75 μm × 20-cm reverse phase column (ReproSil-Pur C18, 3 μm, Dr. Maisch GmbH, Germany) for peptide separation. Analytical column temperature was maintained at 50 ºC by a column oven (Sonation GmBH, Germany). Peptides were eluted with a 3–40% acetonitrile gradient over 60 min at a flow rate of 250 nL min$^{-1}$. The mass spectrometer was operated in data-dependent (DDA) mode with survey scans acquired at a resolution of 120,000 over a scan range of 300–1750 $m/z$. Up to fifteen most abundant precursors from the survey scan were selected with an isolation window of 1.6 Th and fragmented by higher-energy collisional dissociation with Normalized Collision Energies (NCE) of 27. Maximum ion injection time for the survey and MS/MS scans was 60 ms and the ion target value for both scan modes was set to 3e6.

**Data processing**. All mass spectra were first converted to mgf peak list format using Proteome Discoverer 1.4 and the resulting mgf files searched against a human UniProt protein database using Mascot (Matrix Science, London, UK; version 2.5.0; www.matrixscience.com). Decoy protein sequences with reversed sequence were added to the database to allow for the calculation of false discovery rates (FDR). The search parameters were as follows: (i) up to two missed tryptic cleavage sites were allowed; (ii) precursor ion mass tolerance = 10 ppm; (iii) fragment ion mass tolerance = 0.8 Da; and (iv) variable protein modifications were allowed for methionine oxidation, deamidation of asparagine and glutamines, cysteine acrylamide derivatization and protein N-terminal acetylation. MudPit scoring was typically applied using significance threshold score $p < 0.01$. Decoy database search was always activated and, in general, for merged LS-MS/MS analysis of a gel lane with $p < 0.01$, FDR averaged around 1%. Mascot search results were imported into Scaffold (Proteome Software, Inc., Portland, OR; version 4.7.3) to further analyse tandem mass spectrometry (MS/MS) based protein and peptide identifications. X! Tandem (The GPM, thegpm.org; version CYCLONE (2010.12.01.1) was then performed and its results were merged with those from Mascot. The two search engine results were combined and displayed at 1% FDR. Protein and peptide probability was set at 95% with a minimum peptide requirement of one. In each of the Scaffold files that validate and import Mascot searched files, peptide matches, scoring information (Mascot, as well as X! Tandem search scores) for peptide and protein identifications, MS/MS spectra, protein views with sequence coverage and more, can be accessed. To read the Scaffold files, free viewer software can be found at: http://www.proteomesoftware.com/products/free-viewer. Mass spectra files were also analysed using the MaxQuant proteomics data analysis workflow (version 1.6.0.1) with the Andromeda search engine[68]. Raw mass spectrometer files were used to extract peak lists which were searched with the Andromeda search engine against human or mouse proteome and a file containing contaminants such as human keratins. Trypsin specificity with two missed cleavages with the minimum required peptide length was set to be seven amino acids. N-acetylation of protein N-termini, oxidation of methionines and deamidation of asparagines and glutamines were set as variable modifications. For the initial main search, parent peptide masses were allowed mass deviation of 20 ppm. Peptide spectral matches and protein identifications were filtered using a target-decoy approach at a false discovery rate of 1%. We used the raw MS1 intensity and spectral counts for protein quantitation. Quantile normalizations were performed within the replicates of the same sample. Missing values were filled by the minimal raw intensity across all the replicates of the same sample type. All resulting MS1 raw intensity signals were log10 transformed before being subjected to further statistical analyses.

**Bioinformatics analyses**. Differential connectivity (DC) analysis analysis was carried out on the pre-processed raw intensity data or/and MS/MS spectral counts (un-transformed) by performing Student's t-test (two-sided) analyses. Generated $p$-values (Raw.p) and fold changes (Raw.FC) for each identified protein were based on pre-processed raw intensity data and the $p$-values (SC.p) and fold changes (SC.FC) for each identified protein based on MS/MS spectral counts are found in Supplementary Data 1. Fold change was calculated a ratio of mean values from disease sample and from healthy/control samples (Human brains: AD/ND, iPSC derived neurons: APPdp/WT, transgenic mice: PS19/WT, N2a cells: tau-transfected/empty vector (EV)-transfected). For the PPI network visualization (Fig. 3b, upper panel) and iGSEA (Fig. 3c) on human brain samples, a threshold of Raw.p ≤ 0.25 was used to select AD-specific proteins (Raw.FC > 1) and ND-specific (Raw.FC < 1) proteins. For GSEA, the statistical thresholds were set as follow: Human brains: Raw.p ≤ 0.1; iPSC derived neurons: Raw.p ≤ 0.25 or SC.p ≤ 0.25 (SC.FC≠1); transgenic mice: Raw.p ≤ 0.25 or SC.p ≤ 0.25(SC.FC ≠ 1); N2a cells: Raw.p ≤ 0.25. The full list of DE proteins is included in Supplementary Data 5. The heatmap (Fig. 3b, bottom panel) was generated in R using the "gplots" package. Proteins were selected using a threshold of Raw.p ≤ 0.1 and Raw.FC > 1 for AD-specific proteins and a threshold of Raw.p ≤ 0.1 and Raw.FC < 1 for ND-specific proteins. For representation, average fold change was calculated based on log10 transformed raw intensity values of replicates. The synapse biology related gene set was prepared from the SynGO database (updated till 01.2018) (The Synapse Gene Ontology and Annotation Initiative, https://www.syngoportal.org) with exclusion of vague GO terms: "protein binding", "plasma membrane" and "integral component of plasma membrane". SynGO terms were included in Supplementary Data 4. The chaperome member list was created based on the reported 332 human chaperome dataset[69] and is available in Supplementary Data 1. We combined all entries from BioGrid (v.3.4.160) and IntAct (version 05.2018) to create a dataset that inventories all documented human-human (Homo sapiens) PPIs. As a quality control for selecting valid PPIs, interactions annotated as "genetic" (or psi-mi:MI:0208 in IntAct) or those from experiments of "Co-localization", "Genetic interference", "Synthetic Rescue", "Synthetic Growth Defect" and "Synthetic Lethality" were removed. Unless otherwise specified, all PPIs were based on the combined database, prepared as described above. The interactome network in Fig. 3b was built in Cystoscape v3.61 by importing all the proteins identified from the human AD dataset, using the aforementioned PPI database. Spring-electric algorithm was applied to generate the network layout. The node size and node colour represent the log10 transformed $p$-value (Raw.p) and the fold change (Raw.FC) data, respectively, from the DC analysis. However, nodes were

coloured in grey if Raw.p > 0.25 (i.e., were not differentially expressed). For figure clarity, the network edges and node labels were not shown but are available in tables associated with Cytoscape file 1. To investigate the functionality of the chaperome's interactome, we designed the iGSEA method. It takes information from both PPI and Pathway enrichment analyses into analysis. The method applies GSEA within each local interactome of a given chaperome member (through PPI), which summarizes the most prominent biological functions extended from the chaperome member. By this method an additional layer of annotations can be attributed to the chaperome members from the Gene Ontology (GO) terms of their direct interactors. To illustrate the epichaperome-mediated functional rewiring, the iGSEA is performed for each pool containing either AD- or ND-specific proteins as pre-defined by the DC analysis. GO enrichment analysis was performed using R package "ReactomePA", "clusterProfiler" and "org.Hs. eg.db". p-values were adjusted by the method of Benjamini Hochberg (FDR). For visualization, data with adjusted p-value < 0.1 were imported in Cystoscape v3.61 using the columns "Chaperome_ID" (Protein identifier of Chaperome member) and "Description" (GO terms) as interactor A and B. Associated data (p-values, GeneRatio, BgRatio, qvalue and adjusted p-values) were imported as edge attributes. The edges were coloured in blue if the Chaperome-GO association was enriched (adjusted p-value < 0.1) in the ND-specific protein set, or in red, if enriched in the AD-specific protein set (adjusted p-value < 0.1). To generate the figure, the hierarchical layout (yFiles) was applied with edge bends and unconnected GO terms removed. GO terms were drawn in yellow squares and chaperome members were drawn in circles with the node colour representing the fold changes in raw intensity values (Raw.FC). For Fig. 3c, only the SynGO terms were presented. SynGO terms associated with first-degree interactors of each chaperome member were connected by solid lines. The colour of the lines indicates whether the annotations were connected to AD-specific (DC proteins with $FC_{AD/ND} > 1$) or ND-specific interactors (DC proteins with $FC_{AD/ND} < 1$). Full iGSEA data were included in Supplementary Data 2 and associated Cytoscape file 1. For cross-species comparison, proteins of mouse (Mus musculus) origin were converted into human orthologs (see below), and all GSEAs were performed using human-based pathways (from Reactome and Gene Ontology databases). DC proteins were ranked based on protein fold change in decreasing order (Raw.FC) as required by the GSEA algorithm. GSEA was performed in R using the package "ReactomePA" and "clusterProfiler", with the number of permutation (nPerm) set to 10000 and pAjustMethod set to "none". Raw input data (DC proteins) and output data of GSEAs are available in Supplementary Data 5 (input) and Supplementary Data 6 (output) and in Cytoscape file 3 (visualization). The Venn diagram (Fig. 4) was generated using R package "VennDiagram". For Fig. 3d, the Reactome pathway network was visualized in a tree structure based on the hierarchical structure provided by Reactome. Proteins were ranked by decreasing order in fold change (AD over ND: $FC_{AD/ND}$) before subjected to GSEA. Node colour represents the normalized enrichment score (NES) from the core enrichment proteins. A positive score (red) indicates gene set enrichment at the top of the ranked list (high $FC_{AD/ND}$ value); a negative score (grey) indicates gene set enrichment at the bottom of the ranked list (low $FC_{AD/ND}$ value). The border colour represents the p-value of the pathway enrichment from GSEA. The size of the nodes indicates the number (setSize) of DE proteins identified in the pathways. For Fig. 4 (left), the Venn diagram illustrates the overlap in protein identity (DC proteins) across all four samples (human brains, iPSC derived neurons, transgenic mice and N2a cells). For Fig. 4 (right), merged GSEA results across all four samples are illustrated. Only pathways with significant enrichment (p-value < 0.1) were shown. The node size represents the sum of the DC proteins in the pathway that are significantly enriched (p-value < 0.1) in each of the four samples. Pie charts in the nodes illustrate the proportion of the DC proteins in each sample. For Fig. 5e, the Dot plot represents the GSEAs on DC proteins from human brain (AD over ND) samples and tau-transfected N2a cell lines (tau-transfected versus EV-transfected) for a selected panel of Reactome pathways. Normalized enrichment score was indicated by node colour. The number of identified DC proteins (setSize) were indicated by the size of the nodes. All data related to GSEA are available in Supplementary Data 5, 6 and in Cytoscape file 3. Proteins originated from mus musculus (from the PS19 and N2a datasets, see Supplementary Data 1) were converted into human orthologs before being subjected to GSEA (using Gene ontology terms or Reactome pathways). To limit redundancy in protein ortholog mapping, we adopted a step-wise conversion workflow. The first step directly converts the "_MOUSE" suffix of the UniProt entry name of mouse proteins into "_HUMAN". The proteins were retained if the converted UniProt entry name was valid and had a UniProtAC identifier (UniProt accession ID, e.g. "P08238" for human HSP90β). Second, proteins that failed to match in the first step were converted based on the human-mouse ortholog data table from the HCOP (https://www.genenames.org/cgi-bin/hcop). The protein IDs in the data table (for both human and mouse) have been pre-processed by converting Entrez Gene names (in the "human_entrez_gene" and "mouse_entrez_gene" columns) into UniProtAC or by extracting UniProtAC from the "human_assert_ids" or "mouse_assert_ids" columns. Duplicated records and lines with missing UniProtAC were removed. The resulting mouse-human protein mapping table was attached as Supplementary Data 3. Proteins successfully mapped in Step 1 were labelled as "Direct". Proteins mapped in Step 2 using the HCOP table were labelled as "HCOP". For further GSEA, only UniProtAC ID (permanent) was used as the sole protein identifier.

**Statistical analyses.** Statistics were performed and graphs generated using Prism 7 software (GraphPad). Statistical significance was determined using either Student's t-tests or ANOVA. Means and standard errors were reported for all results unless otherwise specified. Effects achieving 95% probability (i.e., p < 0.05) were interpreted as statistically significant. No statistical methods were used to predetermine sample sizes but these are similar to those generally employed in the field. Experiments described in Figs. 2b, 3e, 7b–d, 8b-d and Supplementary Figs. 4–8 were randomized and investigators were blinded to allocation and outcome assessments. All other experiments were not randomized and investigators were not blinded. No samples were excluded from analyses. Details about statistics are reported in the Source Data file.

**Reporting summary.** Further information on research design is available in the Nature Research Reporting Summary linked to this article.

## Data availability
LC-MS data, in total 647 raw files and peak files, that support the findings of this study have been deposited in MassIVE with the MSV000083484 accession number [ftp:// massive.ucsd.edu/MSV000083484/]. The iGSEA code and the Cytoscape files have been deposited in GitHub with the accession code chiosislab/Chaperomics_AD_2019 [https:// github.com/chiosislab/Chaperomics_AD_2019]. Datasets associated with chaperomics analyses are available in the Supplementary Information as Supplementary Data 1 through 6. The source data underlying all main and supplementary figures are provided as a Source Data file.

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

## Acknowledgements

We thank Dr. Jenna Carroll for her advice on the design and the implementation of the behavioural studies, Drs. Virginia Lee, John Trojanowski and Kurt Brunden for the PS19 mice, Suzana Petanceska, Larry Refolo, Gav Pasternak and Howard Fillit for their guidance and support throughout the project, Dr. Peter Davies for his generosity in providing the tau antibodies and sharing with us the staining protocols, the NIH Neurobiobank at the University of Maryland, Baltimore, MD, at the Human Brain and Spinal Fluid Resource Center of VA Greater Los Angeles Healthcare System, CA and at Columbia University, New York, NY for providing the human tissues, Dr. Larry Goldstein for providing the iPSC-derived AD neurons, and Ning Fan, Mesruh Turkekul and Sho Fujisawa for technical support. This work was supported by NIH grants: R56 AG061869, U01 AG032969, P01 AG014449, P01 AG017617, R01 AG043375, R01 CA172546, K01 AG32364, P30 CA008748 (NCI Core Facility Grant), S10 RR027990 and U54 OD020355, and funds from Coins for Alzheimer's Research, ADDF, Appel Alzheimer's Disease Research Institute and the CurePSP Foundation (624-2016-07).

## Author contributions

M.C.I., S.G. and J.K3 designed and performed the behavioural, toxicity and survival studies. S.J. designed and performed the biochemical and functional validation studies. T.W. designed and performed the computational analyses and the data visualization. A.B. designed and performed the epichaperome detection in mice, and M.P.D. and A.C.P. in human patients, by imaging. A.Y.C. and N.D.M.G. designed and performed electrophysiology studies. K.H.A. and S.K. performed iPSC differentiation into neurons and drug treatment studies. H.E.-B. and T.A.N. performed the mass spectrometry sample preparation and protein identification. S.G., A.R., M.U., P.P., A.B., M.R., S.S., A.B., K.X., S.K., J.R.W., J.W.S., P.Y., LY.L.C. and A.G. performed experiments. J.W.S., T.T., W.S., L.S., V.F., S.N., A.A.N. and N.P. provided reagents. W.L., E.deS., S.S., K.M.-T., M.P.D.

and A.C.P. participated in the design and analysis of various experiments, S.P. performed statistical analyses and G.C., S.D.G., S.S. and W.L. wrote the paper.

## Competing interests

Memorial Sloan Kettering Cancer Centre holds the intellectual rights to this portfolio (PCT/US2007/072671 and WO2013/009655). Samus Therapeutics Inc, of which G.C. has partial ownership, and is a member of its scientific advisory board, has licensed PU-AD and PU-AD PET. G.C., T.T., N.P. and W.L. are inventors on the licensed intellectual property. All other authors declare no competing interests.
