## [Peer Review File · Nature Communications]

Reviewers' comments:

Reviewer #1 (Remarks to the Author):

In this revised manuscript, Chiosis et al. have addressed my prior concerns. They clarify that PU-AD specifically binds to Hsp90 incorporated into epichaperome networks. The effects of the compound are intriguing and warrant publication.

Reviewer #2 (Remarks to the Author):

The authors present an interdisciplinary study of the role of interactome alterations to neuronal functioning in Alzheimer's disease, combining analyses of cellular and animal models with results obtained from human specimens. While frequently hypothesized or addressed in isolated, fragmented approaches in individual model systems or at hand of focussed single gene or single mechanism studies, the presented work makes a solid and relevant systems-level integrative approach towards addressing a complex biomolecular setting at relevant scale, and goes a step further, testing therapeutic strategies through targeted pharmacological interventions into the system at play.

It should be stated at the outset, that compared to diverse other studies, this work should be credited for the advance of probing physical interactome changes across models systems, and further rounded up by targeted pharmacological intervention studies with phenotypic in vivo readouts.

The authors start with an analysis of epichaperome expression using the targeted ¹³¹I-PU-AD radiolabelled probe and correlate epichaperome expression with disease progression. Importantly, the authors conclude that epichaperome formation, as a switch from native chaperome expression, precedes tau pathology in the mouse hippocampus, suggesting the mechanistic relevance and prognostic potential of chaperome/epichaperome biology in the onset of AD pathology.

The authors then characterise in biopsies obtained from human postmortem brains epichaperome expression in human brain tissue, followed by confirmation in live patient epichaperome imaging, confirming specificity of epichaperomes to AD-afflicted brains.

Using chemical proteomics through PU-beads, targeting HSP90 to pull-down epi-chaperomes and associated interactomes followed by MS-based detection of interaction partners in diverse mouse, cellular and human samples, followed by network analyses and iGSEA, confirm systems-level network alterations in AD vs. controls, referred to as epichaperome rewiring, and conclude links to synaptic dysfunction in AD through gene ontology enrichment analyses.

The authors conclude that connectivity rewiring from healthy to AD brains, manifested by epichaperome-mediated proteome-wide connectivity changes and functional proteome-wide disturbances in AD, entail epichaperome-mediated defects in synaptic plasticity, cell signaling, translation, cell cycle regulation, axonic guidance and others, which the authors confirm by assessing key effector proteins in pathways of importance to synaptic fitness, such as Rho GTPase signaling in actin remodelling and cytoskeletal reorganization, as represented by p-cofilin. The authors hypothesize links to ensuing tau-induction and tau-associated stress and confirm that PU-AD mediated epichaperome repression reset the system to pre-tau states in a tau overexpression model (Fig. 4). Epichaperomes are therefore proposed as a mechanistic trigger of tau-induced functional imbalances in synaptic pathways. Importantly, the authors control and exclude HSF-1 from the MoA, as opposed to GA inhibition of HSP90. Overall, the data thereby provide convincing evidence on a mechanistic link between dynamic protein interactome network remodelling in AD and structural chaperome alterations into epichaperomes. In vivo PET-CT studies in PS19 mice linked epichaperomes to spatial memory impairment and showed improved phenotypes upon

epichaperome targeting with the PU-AD inhibitor, which the authors link to changes in synaptic transmission as resulting from a correction of PCBD (or 'protein connectivity based degeneration') and synaptic protein network rebalancing. Long-term in vivo toxicity studies in mice showed that extended administration of PU-AD inhibitor was non-toxic in C57/BL6 mice and extending survival of PS19 mice by delaying their tau-induced paralysis phenotype.

Overall the study is important and of sufficiently strong and convincing novelty as such that it provides an important contribution to our understanding of the role of proteostasis in Alzheimer's disease. Especially the elegant assessment of chaperome alterations, linking to the previously published concept of the "epi-chaperome" in cancers and neurodegenerative disease, makes this paper a relevant contribution of wide interest, beyond the neurosciences. The authors refer to the inter dependencies between observed chaperome re-modelling and neurodegenerative disease as PCBD, thereby providing a very nice example of the role of the 'edgotype', or the wider interactome, in genotype-phenotype relationships and test hypothesis for pharmacological re-adjustment of the network-based target structures, or "mal-adaptive epi-chaperomes". This work shall serve as an inspiration to other areas of disease relevance as regards the consideration of an understanding of the chaperome and its alterations, within the wider interactome context, to the molecular underpinnings of disease.

Congratulations.

The reviewer would like to state the following few points:

Major points:

1. Ad Fig. 1 - How do the authors justify "epichaperome" expression to be adequately assessed in brain samples solely by the targeted 131 I-PU-AD probe (targeting HSP90)? Please discuss.
2. Other probes analogous to 131 I-PU-AD should be tested in parallel to confirm the effects seen by targeting the 'epi-chaperomes' with HSP90-targeting reagents such as PU-AD or PU-H71.
3. Similarly in human biopsies, as in 1., ad Fig. 2, additional targets beyond HSP90, HSC70, HOP and CDC37 should be tested in order to make an adequate statement about heterogenic chaperomes that are comprised of hundreds of members.
4. Can epichaperome collapse be observed when epichaperomes are targeted via other direct polypeptide targets than HSP90, using other probes than PU-AD or PU-H71? The underlined difference between epichaperome-targeting PU-AD or PU-H71 vs. conventional HSP90 inhibitors could be corroborated further by the direct comparison to some of those. Please comment and emphasize supporting evidence.
5. The authors have probed the chaperome, and have detected and correlated chaperome connectivity changes, termed maladaptive epi-chaperomes, with AD. Have the authors assessed the remainder, or parts, of the non-chaperome interactome in a systematic, targeted manner? Do we know whether unrelated, or related processes are at play that could contribute, trigger or modulate the phenotypes related here to chaperome to epichaperome transitions? These would support the approach, rationalised by the authors as 'chaperomics' in the context of systems-level alterations. Please discuss.

Minor points:

1. The terms chaperone and chaperome should be more clearly used to distinguish as chaperones, individual proteins, while the chaperome should be used only to refer to the ensemble of the human chaperome.
2. The legend to Fig. 2 should indicated "Chaperones", not "Chaperomes", as the chaperome should refer to the ensemble of all chaperones, not single protein nodes in the network.

Upon commenting and adding to the discussion in response to the above points, this manuscript should be of interest to the readership of Nature Communications and surely will make a contribution towards the void in significant clinical advancement in curing AD.

Overall, I would like to congratulate the authors on having achieved a fundamental contribution on the role of chaperome alterations in the context of the cellular interactome network in neurodegenerative disease that should trigger further productive initiatives in proteostasis neurosciences through raising awareness to the important field of proteostasis in the area of neurosciences.

Dr. Marc Brehme Vienna, 20th October 2019

We thank reviewers for their insightful suggestions and detailed review of our manuscript. We are delighted that Reviewer #1 has found all prior concerns satisfactorily addressed and recommends publication. We are very pleased with the insightful summary provided by Reviewer#2 which excellently captures the essence of this manuscript.

Reviewer# 2 recommends that we add to the Discussion our responses to the points listed below. We agree these strengthen the manuscript and we thank Reviewer#2 for suggesting them.

1. *Ad Fig. 1 - How do the authors justify “epichaperome” expression to be adequately assessed in brain samples solely by the targeted 131 I-PU-AD probe (targeting HSP90)? Please discuss.*
2. *Other probes analogous to 131 I-PU-AD should be tested in parallel to confirm the effects seen by targeting the ‘epichaperomes’ with HSP90-targeting reagents such as PU-AD or PU-H71.*
3. *Similarly in human biopsies, as in 1., ad Fig. 2, additional targets beyond HSP90, HSC70, HOP and CDC37 should be tested in order to make an adequate statement about heterogenic chaperomes that are comprised of hundreds of members.*
4. *Can epichaperome collapse be observed when epichaperomes are targeted via other direct polypeptide targets than HSP90, using other probes than PU-AD or PU-H71? The underlined difference between epichaperome-targeting PU-AD or PU-H71 vs. conventional HSP90 inhibitors could be corroborated further by the direct comparison to some of those.*
Please comment and emphasize supporting evidence.

Response. These are all excellent interrelated comments that we address collectively. Opportunities in selectively inhibiting the epichaperome over more abundant cellular pools of chaperones, arise from the nature of the epichaperome itself, which presents as stable and multi-partner complexes formed by strong interactions, as opposed to other cellular forms of chaperones present in dynamic complexes of weak interactions and with a limited number of partners (Rodina et al., Nature, 2016; Ref #17, Joshi et al., Nature Reviews Cancer, 2018; Ref #18, Wang et al., JBC, 2019; Ref #20, Taldone et al., CSH Perspectives Biology, 2019; Ref #24). These structural and dynamic features provide an approach for the specific binding of small molecules (such as PU-H71 and PU-AD) through kinetic selectivity to differentiate HSP90 residing in the epichaperome from the more abundant HSP90 pools (Rodina et al., Nature, 2016; Ref #17, Wang et al., JBC, 2019; Ref #20, Taldone et al., CSH Perspectives Biology, 2019; Ref #24).

This is a new avenue for rational therapeutics aimed at correcting PCBD through de-connecting aberrant epichaperome structures and the proteome-wide pathways they rewire. It will therefore be interesting, and certainly useful from a therapeutic perspective, to investigate whether modulation of other hubs in the epichaperome network (such as of HSC70 and other candidates) may induce, or not, epichaperome collapse, and in turn correct PCBD.

Radiolabeled forms of PU-H71 and PU-AD, are epichaperome detection reagents, akin to how an antibody can recognize a specific protein, as they dissociate from HSP90 residing in epichaperomes much more slowly (i.e., over several hours to days) than they do from other HSP90 pools (i.e., minutes to few hours); this difference in the k_{off} (i.e. dissociation constant) provides it with epichaperome selectivity (Rodina et al., Nature, 2016, Ref #17, Wang et al., JBC, 2019; Ref #20, Pillarsetty et al., Cancer Cell, 2019; Ref #25). These probes therefore are validated and can be used to detect and quantify epichaperome positivity, as evidenced in Figure 1. In fact, these reagents are used in clinic to detect epichaperome positivity by positron emission tomography (PET) imaging in human patients (NCT01269593, NCT03371420, and Pillarsetty et al., Cancer Cell 2019; Ref #25).

Accordingly, we added a summary of the above paragraphs in this response letter to the Discussion (in blue letters).

In Figure 2 we provide analysis on several ‘core’ epichaperome network components. Analyses by native PAGE/immunoblotting are limited to those proteins for which we have identified antibodies that can recognize the native conformation/complexation adopted by each such chaperone when integrated in the epichaperome networks. Identification of additional antibodies, suitable for detection of other epichaperome components, is a project in itself as will require the purchase of dozens of commercial antibodies for each chaperone, and then the testing of such antibodies for their ability to detect the specific conformer/complex. However, we argue we

have already performed the identification of the hundreds of epichaperome members, because mass spectrometry on the chaperomics cargo, which we perform next in the manuscript (Figure 3 and onwards), provides the identity of all such chaperome members, in a large-scale unbiased manner.

A systematic comparison between PU-AD or PU-H71 and conventional HSP90 inhibitors, beyond the data we provide in Figure 6C, is outside the scope of the present manuscript. We agree with Reviewer#2 this is an important topic, and hope our studies will raise awareness on the heterogeneity of the chaperome in disease, and in turn, on its implication in inhibitor discovery and development. We have published several review articles over the last two years detailing on the topic, and these references are incorporated in the manuscript (Joshi et al., Nature Reviews Cancer 2018; Ref #18, Taldone et al., CSH Perspectives Biology 2019; Ref #24, Wang et al., JBC 2019; Ref #20).

5. The authors have probed the chaperome, and have detected and correlated chaperome connectivity changes, termed maladaptive epi-chaperomes, with AD. Have the authors assessed the remainder, or parts, of the non-chaperome interactome in a systematic, targeted manner? Do we know whether unrelated, or related processes are at play that could contribute, trigger or modulate the phenotypes related here to chaperome to epichaperome transitions? These would support the approach, rationalised by the authors as 'chaperomics' in the context of systems-level alterations.

Please discuss.

Response. These are intriguing questions and we thank the Reviewer for curiosity on the subject.

Currently, the AD research community is facing many different challenges in elucidating the potential functions of gene variants and environmental stressors in the onset and progression of AD. Our data provide a confluence of evidence from cellular model, animal model, live human, and postmortem human brain tissues that chaperomics has the ability to discover the identity of proteome-wide alterations induced by individual stressors. Chaperomics therefore may enable many novel inquiries into AD biology, from screening for changes in protein function, PPIs, protein complexes; screening genotype-to-phenotype effects; discovery of new targets for AD treatment; and classification of AD based on functional proteome signature, among others. Conversely, current 'omics' tools are limited in their ability to pinpoint specific protein activity and functions, which are the end-game mediators of the effects of AD risk gene variants, and to define the molecular alterations triggered by a combination of genetic and environmental factors.

In the progression to AD, stressors and vulnerabilities such as aging, genetics, environmental factors, and others drive changes in the brain that occur and accumulate over decades. These result in imbalances in the connectivity of the neuronal circuitry, but also intracellularly in the connectivity of neuronal proteins and protein pathways. An investigation through the use of chaperomics in a linear fashion: *i*) region, *ii*) disease state, *iii*) neuropathology, *iv*) cognition, and *v*) sex into the dysregulation of PCBD from no cognitive impairment, to MCI to AD using postmortem tissues may therefore provide the causal and functional link between stressors and proteome dysfunction as it manifests during AD progression.

Minor points:

- 1. The terms chaperone and chaperome should be more clearly used to distinguish as chaperones*
- 2. The legend to Fig. 2 should indicated "Chaperones", not "Chaperomes", as the chaperome should refer to the ensemble of all chaperones, not single protein nodes in the network.*

Response. We thank the Reviewer for the thorough review and we went through the text in the body of the manuscript and Figure Legends carefully and edited as appropriate.

REVIEWERS' COMMENTS:

Reviewer #2 (Remarks to the Author):

Dear Prof. Chiosis and co-workers,

Congratulations once more on presenting important insights in a convincing manner in this manuscript.

The responses provided clearly and thoroughly addressed all comments raised throughout and the update to the manuscript, including additional discussion points inserted, round up the manuscript from a general readers' point of view. The discussion now sufficiently addresses those points, including potential risks and limitations that need to be addressed in future work.

Thank you for this beautiful work. I am very pleased to recommend this manuscript for publication in Nature COMMUNICATIONS and look forward to reading it again in print.

Best regards,
Marc Brehme, Ph.D.
Vienna, 17th Nov 2019

REVIEWERS' COMMENTS:

Reviewer #2 (Remarks to the Author):

Dear Prof. Chiosis and co-workers,

Congratulations once more on presenting important insights in a convincing manner in this manuscript.

The responses provided clearly and thoroughly addressed all comments raised throughout and the update to the manuscript, including additional discussion points inserted, round up the manuscript from a general readers' point of view. The discussion now sufficiently addresses those points, including potential risks and limitations that need to be addressed in future work.

Thank you for this beautiful work. I am very pleased to recommend this manuscript for publication in Nature COMMUNICATIONS and look forward to reading it again in print.

Response: Thank you. We are grateful for the positive comments on the resubmission and appreciate the Reviewer concurs with our assessment of the importance of these novel findings of PCBD in the context of AD.